# Parametrized Power-Iteration Clustering for Directed Graphs

Gwendal Debaussart-Joniec [† 1]   Harry Sevi [† 1]   Matthieu Jonckheere [2]   Argyris Kalogeratos [1]

## Abstract

Vertex-level clustering for directed graphs (digraphs) remains challenging as edge directionality breaks the key assumptions underlying popular spectral methods, which also incur the overhead of eigen-decomposition. This paper proposes *Parametrized Power-Iteration Clustering* (ParPIC), a random-walk-based clustering method for weakly connected digraphs. ParPIC builds on the Power-Iteration Clustering paradigm, which uses the rows of the iterated diffusion operator as a data embedding, and has three important features: the use of parametrized reversible random walk operators, the automatic tuning of the diffusion time, and the efficient truncation of the final embedding, which produces low-dimensional data representations and reduces complexity. Empirical results on synthetic and real-world graphs demonstrate that ParPIC achieves competitive clustering accuracy with improved scalability relative to spectral and teleportation-based methods.

## 1. Introduction

Random-walk- and diffusion-based methods are central to graph representation learning, with impact across clustering, dimensionality reduction, and network analysis. For undirected graphs, there is a mature theoretical foundation: the natural random walk on the graph is reversible with a (generally) unique stationary distribution, its spectrum is real-valued, and the induced diffusion geometry is widely studied, supporting the notions of distance and embedding (Coifman et al., 2005; Nadler et al., 2006a; Coifman & Lafon, 2006; Shan & Daubechies, 2022). Power-iteration schemes (Lin & Cohen, 2010; Liu et al., 2021; Ye et al., 2016) exploit this structure to efficiently extract multiscale geometric features without using explicit eigen-decomposition.

[†]Equal contribution [1]Université Paris-Saclay, ENS Paris-Saclay, Centre Borelli, CNRS, France. [2]CNRS, LAAS, France. Correspondence to: Gwendal Debaussart-Joniec <gwendal.debaussart@ens-paris-saclay.fr>, Harry Sevi <harry.sevi@protonmail.com>.

*Proceedings of the 43rd International Conference on Machine Learning*, Seoul, South Korea. PMLR 306, 2026. Copyright 2026 by the author(s).

Extending diffusion geometry to directed graphs (digraphs) is non-trivial. Natural random walks on digraphs are generally non-reversible, may be reducible, and their associated operators often admit complex-valued eigenvectors. This complicates both computation and interpretation (Levin & Peres, 2017; Seabrook & Wiskott, 2023). In addition, many digraphs encountered in practice, including some $k$-nearest neighbor graphs built from datapoints, are only weakly connected and hence violate the strong connectivity assumptions underpinning both classical spectral constructions and power-iteration schemes. Common remedies include graph symmetrization (Von Luxburg, 2007; Satuluri & Parthasarathy, 2011) or teleportation-based random walks, such as PageRank (Page et al., 1999; Tabrizi et al., 2013), which enforce ergodicity but alter the original dynamics and may obscure directional information. Alternative Hermitian-based approaches (Cucuringu et al., 2020; Laenen & Sun, 2020; Mohar, 2020) preserve directionality and recover real spectra but use complex-valued entries, hence lacking a probabilistic interpretation. In addition, it is important to distinguish that those approaches have a flow-based cluster definition, which is fundamentally different from the edge-density-based cluster structure that is more standard in the literature, and notably what random-walk dynamics aim at recovering.

Reversible operator constructions based on stationary distributions (Chung, 2005), or vertex measures (Sevi et al., 2025), have emerged as principled ways to retain directionality while recovering diffusion structure, and have established a sound theoretical foundation for diffusion processes on weakly connected digraphs. Such approaches rely on the spectral clustering pipeline: the spectral embedding of the data is produced by the eigen-decomposition of a given Laplacian operator, on which clustering is performed, using methods such as $k$-means. Power-Iteration Clustering (PIC) (Lin & Cohen, 2010; Liu et al., 2021) is an alternative pipeline that clusters directly the rows of an iterated random walk operator, thus avoiding the eigen-decomposition.

***Contributions.*** This paper proposes *Parametrized Power-Iteration Clustering* (ParPIC), a framework that uses power-iterations of a parametrized reversible operator (Fig. 1). This eigen-free approach extends the spirit of PIC to weakly connected digraphs, while offering computational efficiency:

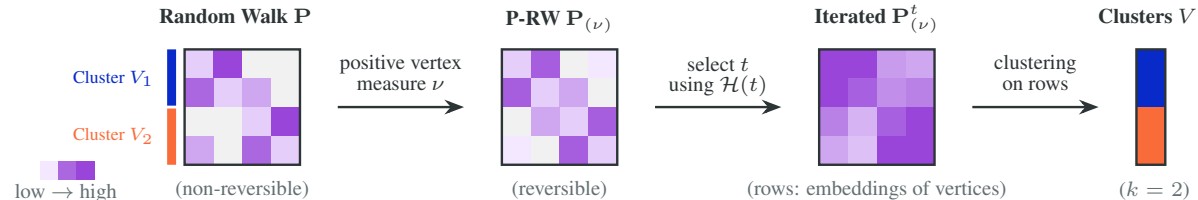

*Figure 1.* **The ParPIC pipeline.** Given a digraph $\mathcal{G}$ with natural random walk $\mathbf{P}$, a parametrized reversible random walk operator $\mathbf{P}_{(\nu)}$ is constructed based on a vertex measure $\nu$ (Sec. 3). Power-iterations of $\mathbf{P}_{(\nu)}$ are then performed to compute $\mathbf{P}^t_{(\nu)}$ at a selected diffusion time $t$ (or an approximation to $\mathbf{P}^t_{(\nu)}$ is computed, Sec. 3.4). The final data partition is produced by clustering the rows of $\mathbf{P}^t_{(\nu)}$, e.g. using $k$-means. This process avoids explicit eigen-decomposition while preserving directional diffusion dynamics for effective clustering.

- **Eigen-free diffusion clustering for digraphs.** ParPIC is a power-iteration-based clustering method that extends diffusion and PIC-style algorithms to digraphs without the need for eigen-decomposition. This brings a significant computational advantage, which enables scalability to larger graphs.
- **Parametrized reversible diffusion operators.** Building on recent advances on reversible random walks for digraphs, this work shows how parametrized vertex measures induce well-defined diffusion dynamics suitable for clustering.
- **Unsupervised diffusion time selection.** An entropic criterion is proposed for selecting diffusion scales directly from the power-iteration, remaining eigen-free and computationally efficient.
- **Empirical validation.** Experiments on synthetic and real-world digraphs demonstrate competitive clustering performance at reduced computational cost.

## 2. Background

This section introduces the notations, reviews diffusion geometry for undirected graphs, and discusses the limitations of diffusion geometry and spectral methods on digraphs.

### 2.1. Notations

Let $\mathcal{G} = (V, E, w)$ be a weighted digraph with $N = |V|$ vertices. Each edge $(i, j) \in E$ has weight $w(i, j) \geq 0$ representing influence from $i$ to $j$. The adjacency matrix $\mathbf{W} \in \mathbb{R}^{N \times N}$ has entries $\mathbf{W}_{ij} = w(i, j)$ (written as $\mathbf{W}_{ij}$ or $\mathbf{W}(i, j)$, interchangeably). The out-degree and in-degree of vertex $i$ are $d_{\text{out}}(i) = \sum_j \mathbf{W}_{ij}$ and $d_{\text{in}}(i) = \sum_j \mathbf{W}_{ji}$, with corresponding diagonal matrices $\mathbf{D}_{\text{out}} = \text{diag}(d_{\text{out}}(1), ..., d_{\text{out}}(N))$ and $\mathbf{D}_{\text{in}} = \text{diag}(d_{\text{in}}(1), ..., d_{\text{in}}(N))$. A strictly positive function $\nu : V \to \mathbb{R}_+$ is a *vertex measure*, represented as vector $\nu \in \mathbb{R}^N_+$ (written as $\nu(i)$ or $\nu_i$, interchangeably), with associated diagonal matrix $\mathbf{D}_\nu$ where $(\mathbf{D}_\nu)_{ii} = \nu(i)$.

A random walk on $\mathcal{G}$ is a Markov chain whose transition probabilities follow the *outgoing* edge structure (Brémaud, 2013). The transition matrix of the natural random walk is defined as $\mathbf{P} = \mathbf{D}_{\text{out}}^{-1}\mathbf{W}$. The terminology "$\mathbf{P}$ is irreducible" or "$\mathbf{P}$ is reversible" is used as shorthand for saying that the Markov chain has these properties. A stationary distribution is a vertex measure $\pi$ satisfying $\pi\mathbf{P} = \pi$. A Markov chain is *irreducible* if every state is reachable from every other state, i.e. if for every pair $(i, j)$ there exists a $t$ such that $\mathbf{P}^t_{ij} > 0$. It is *reversible* if it satisfies $\pi(i)\mathbf{P}(i, j) = \pi(j)\mathbf{P}(j, i)$ for all $i, j$.

### 2.2. Diffusion Geometry for Undirected Graphs

Diffusion geometry (Coifman et al., 2005; Coifman & Lafon, 2006) builds upon the random walk operator on an *undirected* graph to define a multiscale geometric framework for data analysis. Throughout this subsection, $\mathcal{G}$ is assumed to be undirected. Two concepts are central to diffusion geometry: *diffusion distances* and *diffusion maps*. The diffusion distance at time $t$ between two datapoints is:

$$\mathcal{D}^2_t(i, j) = \sum_{k=1}^{N} \frac{1}{\pi(k)} \left( \mathbf{P}^t_{i,k} - \mathbf{P}^t_{j,k} \right)^2. \qquad (1)$$

Intuitively, small diffusion distance indicates that vertices behave similarly under the random walk viewpoint, and thus likely belong to the same cluster. The time parameter $t$ controls the scale at which the structure is examined. The diffusion map at time $t$ is defined by the spectral embedding of the iterated random walk operator $\mathbf{P}^t$. Formally, if $\mathbf{P} = \sum_{i=1}^{N} \lambda_i \phi_i \phi_i^\mathsf{T}$ is the eigen-decomposition of $\mathbf{P}$, the diffusion map at time $t$ at point $i$ is given by:

$$\mathbf{\Psi}_t(i) = \left( \lambda_1^t \phi_1(i), \lambda_2^t \phi_2(i), ..., \lambda_N^t \phi_N(i) \right)^\mathsf{T}, \qquad (2)$$

which embeds vertices in $\mathbb{R}^N$. For practical purposes, one can keep only the $d$ largest eigenvalues to obtain a low-dimensional embedding in $\mathbb{R}^d$. This embedding has the property that diffusion distances can be directly computed as Euclidean distances in the diffusion map space, i.e. $\mathcal{D}^2_t(i, j) = \left\| \mathbf{\Psi}_t(i) - \mathbf{\Psi}_t(j) \right\|_2^2$. Essentially, the diffusion map embedding captures the connectivity structure of the graph at a scale that grows with $t$.

## 2.3. Limitations of Diffusion Geometry and Spectral Methods on Digraphs

Extending diffusion geometry to digraphs is not straightforward. In particular, classical diffusion geometry assumes that the random walk operator is reversible, which guarantees a unique stationary distribution and a real eigen-basis necessary to define the associated diffusion map. However, as discussed earlier, this assumption typically fails on digraphs. As a result, diffusion distances may be ill-defined and spectral embeddings may be unstable or undefined, meaning that both clustering and embedding quality may degrade significantly. These limitations apply to both spectral and power-iteration methods that rely on the random walk operator, as the non-uniqueness of the ergodic law causes issues in the convergence of the transition operator. In summary, applying diffusion geometry on digraphs faces several limitations that motivate the investigation of an operator that is well-defined for any digraph, reversible (in a suitable weighted space), and ergodic under mild assumptions, while preserving directionality information. The next section introduces such an operator based on recent advances in Laplacian definition for digraphs, and more specifically the work of Sevi et al. (2025).

# 3. Parametrized Power-Iteration Clustering

This section presents the Parametrized Power-Iteration Clustering (ParPIC) framework, which relies on a random walk operator parametrized by a vertex measure. Discussion includes the induced diffusion geometry, vertex measure designs, diffusion time selection, and efficient algorithmic deployment.

## 3.1. Parametrized Random Walk Operator

To address the limitations of diffusion geometry on digraphs discussed in Sec. 2.3, a parametrized random walk operator is introduced that generalizes the natural random walk by incorporating a vertex measure $\nu$. This operator is reversible and ergodic under mild conditions while preserving directionality, allowing the extension of diffusion geometry to digraphs in a principled manner.

**Definition 3.1** (Parametrized random walk (P-RW) operator). Let $\mathbf{P} \in \mathbb{R}^{N \times N}$ be a transition matrix, $\nu$ be an arbitrary vertex measure on $\mathbb{R}_+^N$, and define $\xi = \mathbf{P}^\mathsf{T}\nu$. The parametrized random walk operator $\mathbf{P}_{(\nu)}$ is defined as:

$$\mathbf{P}_{(\nu)} = (\mathbf{D}_\nu + \mathbf{D}_\xi)^{-1}(\mathbf{D}_\nu\mathbf{P} + \mathbf{P}^\mathsf{T}\mathbf{D}_\nu), \qquad (3)$$

where $\mathbf{D}_\nu = \mathrm{diag}(\nu)$ (i.e. $(\mathbf{D}_\nu)_{ii} = \nu(i)$), $\mathbf{D}_\xi = \mathrm{diag}(\xi)$.

The P-RW operator $\mathbf{P}_{(\nu)}$ can be interpreted as the transition matrix of a modified random walk on the graph $\mathcal{G}$, whose dynamics are influenced by the vertex measure $\nu$. Proposition 3.2, which follows, shows how the choice of the vertex

measure $\nu$ impacts the random walk dynamics. This operator is inspired by recent works on Laplacian definitions for digraphs using vertex measures (Sevi et al., 2025), and can be seen as a generalization of the random walk operator defined in Chung (2005), which corresponds to the special case where $\nu$ is chosen to be the stationary distribution of $\mathbf{P}$. The proposed approach differs in that it allows a diffusion interpretation due to its normalization, and does not require the vertex measure to be a probability distribution. The flexibility in the choice of $\nu$ allows us to tailor the diffusion dynamics to specific applications by selecting appropriate vertex measures.

**Proposition 3.2** (Impact of $\nu$ on the P-RW operator). *The following statements hold:*

- *The P-RW operator is reversible with respect to the measure $\nu + \xi$.*
- *If the underlying graph $\mathcal{G}$ is weakly connected, $\mathbf{P}_{(\nu)}$ is irreducible.*
- *If the underlying undirected graph $\mathcal{G}$ is aperiodic, then $\mathbf{P}_{(\nu)}$ is aperiodic.*
- *The operator $\mathbf{P}_{(\nu)}$ is continuous with respect to $\nu$.*
- *For ergodic undirected graphs, choosing $\nu = \pi$ recovers the natural random walk, i.e. $\mathbf{P}_{(\pi)} = \mathbf{P}$.*

*Under these conditions, $\mathbf{P}_{(\nu)}$ is ergodic, with $\pi_{(\nu)}$ being its unique stationary distribution, i.e. $\pi_{(\nu)}^\mathsf{T}\mathbf{P}_{(\nu)} = \pi_{(\nu)}^\mathsf{T}$ (Seabrook & Wiskott, 2023).*

This proposition shows that by choosing a strictly positive vertex measure $\nu$, we can ensure that the P-RW operator $\mathbf{P}_{(\nu)}$ is aperiodic and irreducible; moreover it is reversible with respect to the measure $\nu + \xi$ (proportional to $\pi_{(\nu)}$). This is crucial in the definition of diffusion geometry on digraphs, as it allows us to extend the diffusion distance and the associated diffusion kernel to digraphs. The proof is deferred to Appendix E and relies on standard properties for self-adjoint operators and linear algebra. Note also that the choice of $\nu$ directly impacts the stationary distribution $\pi_{(\nu)}$ of the P-RW operator $\mathbf{P}_{(\nu)}$, and hence the geometry induced by the diffusion process. This flexibility allows us to tailor the diffusion dynamics to specific applications via the selection of the vertex measure.

## 3.2. Parametrized Diffusion Geometry for Digraphs

Building on the P-RW operator $\mathbf{P}_{(\nu)}$, a diffusion geometry on digraphs can be defined, generalizing the classical diffusion geometry from Sec. 2.2 to the directed setting. The key innovation is to incorporate the vertex measure $\nu$ in order to preserve directionality while maintaining reversibility. The parametrized diffusion distance is defined to capture graph geometry at scale $t$ while respecting the structural information encoded in $\nu$.

**Definition 3.3** (Parametrized diffusion distance). Let $\mathbf{P}_{(\nu)}$

be a P-RW operator with stationary distribution $\pi_{(\nu)}$, based on a random walk $\mathbf{P}$, and let $\xi = \mathbf{P}^{\mathsf{T}}\nu$. The parametrized diffusion distance at time $t$ between two vertices $i, j \in V$ is defined as:

$$\mathcal{D}^2_{t,(\nu)}(i,j) = \sum_{k=1}^{N} \frac{1}{\pi_{(\nu)}(k)} \left( \mathbf{P}^t_{(\nu)}(i,k) - \mathbf{P}^t_{(\nu)}(j,k) \right)^2, \quad (4)$$

where $\pi_{(\nu)}$ is the stationary distribution of $\mathbf{P}_{(\nu)}$.

This distance metric generalizes the classical diffusion distance to digraphs while preserving key geometric properties. Notably, it captures connectivity structure at scale $t$ while remaining sensitive to the vertex measure $\nu$, which encodes directional information. Unlike classical diffusion geometry, which requires reversibility to be well-defined, our construction explicitly maintains reversibility through the operator definition, enabling diffusion analysis on weakly connected directed graphs. Next, we define the parametrized diffusion map, which provides a spectral embedding of vertices based on the P-RW operator $\mathbf{P}_{(\nu)}$.

**Definition 3.4** (Parametrized diffusion map)**.** Let $\mathbf{P}_{(\nu)}$ be a P-RW operator with stationary distribution $\pi_{(\nu)}$ based on a random walk $\mathbf{P}$, and $\mathbf{P}_{(\nu)} = \mathbf{\Phi}_{(\nu)} \mathbf{\Lambda}_{(\nu)} \mathbf{\Phi}_{(\nu)}^{-1}$ its eigendecomposition. The parametrized diffusion map at time $t$ is defined by:

$$\Psi_{t,(\nu)}(i) = \delta_i^{\mathsf{T}} \mathbf{\Phi}_{(\nu)} \mathbf{\Lambda}_{(\nu)}^t. \quad (5)$$

This definition generalizes the classical diffusion map to digraphs, while preserving spectral embedding properties (Appendix E). Although we avoid eigen-decomposition in practice, this definition characterizes the link between the iteration of the P-RW operator and the diffusion map. Formally, the parametrized diffusion map consists in embedding vertices using the $k$ (as many as the number of clusters) largest eigenvalues of the P-RW operator $\mathbf{P}_{(\nu)}$, scaled by $t$. This embedding captures the connectivity structure of the digraph at scale $t$, while being influenced by the vertex measure $\nu$. Correspondingly, iterating the P-RW operator $\mathbf{P}_{(\nu)}$ essentially applies a smooth function $g_t(\lambda) = \lambda^t$ to the eigenvalues of $\mathbf{P}_{(\nu)}$. As $t$ increases, $g_t$ becomes smoother, preserving the eigenvectors of $\mathbf{P}_{(\nu)}$ associated with the largest eigenvalues. Essentially, this approach can be seen as a smooth alternative to traditional spectral embedding, where the diffusion time $t$ controls the scale at which the structure is revealed.

### 3.3. Designs for the Vertex Measure

There can be many possible choices for the vertex measure $\nu$, which can also be tailored to a specific application. A natural choice is the convex combination of the in-degree and out-degree of each vertex:

$$\nu_\gamma = \gamma d_{\text{in}} + (1-\gamma) d_{\text{out}}, \quad \gamma \in [0,1]. \quad (6)$$

Generally, both in- and out-degrees are considered to be normalized, i.e. $\sum_{i=1}^{N} d_{\text{in}}(i) = \sum_{i=1}^{N} d_{\text{out}}(i) = 1$, so that $\nu_\gamma$ sums to 1; this is not mandatory but can address a potentially significant difference between in- and out-degrees in some digraphs. By adjusting the parameter $\gamma$, the influence of incoming (for $\gamma$ close to 1) or the outgoing connections (for $\gamma$ close to 0) can be emphasized, or those two factors can be balanced. This flexibility is particularly useful in digraphs where the roles of incoming and outgoing edges may differ significantly. In applications such as citation networks or web graphs, the in-degree may reflect popularity or authority, while the out-degree may indicate activity or influence. By tuning $\gamma$, the diffusion process can be adapted to better capture the relevant dynamics for clustering or embedding tasks. In cases where sinks or sources are present in the digraph, setting $\gamma$ to the extremes (0 or 1) can cause the measure to violate the hypotheses of Proposition 3.2. Appendix D.1 provides a comprehensive sensitivity analysis of $\gamma$, showing that $\gamma = 0.5$ (equal weighting) performs robustly across most datasets, while directed structures benefit from other parameter values.

### 3.4. Setting the Diffusion Time

The selection of the time parameter in the diffusion setting is a challenging problem (Shan & Daubechies, 2022; Maggioni & Murphy, 2019; Sevi et al., 2025; Nadler et al., 2006b). In the context of data clustering, there is interest in estimating the time horizon that best reveals the $k$-cluster structure. Recent works (Debaussart-Joniec & Kalogeratos, 2025; Kuchroo et al., 2021) have shown that entropic criteria over the spectrum of the operator can be effective in identifying meaningful scales in diffusion processes. Despite being insightful and inspiring, those measures still rely on the eigen-decomposition of the diffusion operator, which is not compatible with our eigen-free approach.

To assess the effect of diffusion time, an entropic criterion is defined over the rows of the iterated P-RW operator $\mathbf{P}^t_{(\nu)}$. More specifically, the *global row-wise operator entropy* is $\mathcal{H}(t) = \sum_{i=1}^{N} \mathcal{H}_i(t)$, which is defined as the sum of the *row entropies* $\mathcal{H}_i(t)$:

$$\mathcal{H}_i(t) = -\sum_{j=1}^{N} \mathbf{P}^t_{(\nu)}(i,j) \log(\mathbf{P}^t_{(\nu)}(i,j)). \quad (7)$$

The measure $\mathcal{H}(t)$ captures the dynamics of the random walk at time $t$. Intuitively, short random walks do not explore enough of the graph, leading to low entropy. Conversely, longer random walks converge to the stationary distribution of the Markov chain, which results in high entropy. Empirically, this means that a proper intermediate $t$ value should be sought for a given task. The proposed measure satisfies the following properties.

**Proposition 3.5** (Behavior of $\mathcal{H}(t)$)**.** *Let $\mathbf{P}_{(\nu)}$ be a P-RW*

*operator. The row-wise operator entropy $\mathcal{H}(t)$ satisfies:*

- $\mathcal{H}(t)$ *is non-decreasing with $t$;*
- $\lim_{t\to\infty} \mathcal{H}(t) = C(\pi_{(\nu)})$, *where $C(\pi_{(\nu)})$ depends on the stationary distribution. For a multi-component graph, $C(\pi_{(\nu)})$ depends on the stationary distribution of each component.*

The proof is deferred to Appendix E and relies on properties of stochastic matrices and the convergence of Markov chains. By analyzing the behavior of $\mathcal{H}(t)$ as a function of $t$, a diffusion time can be identified that balances exploration and convergence, thereby revealing meaningful cluster structures. To do so, $t$ is selected as the elbow of the curve $t \mapsto \mathcal{H}(t)$ (Fig. 8, in Appendix D.2). As stated earlier, this formulation is eigen-free, relying only on the iterated operator $\mathbf{P}_{(\nu)}^t$. A comprehensive time selection analysis is provided in Appendix D.2, including sampling efficiency validation (Fig. 8) and comparison with fixed time baselines (Fig. 7).

### 3.5. Practical Implementation

This section outlines how the ParPIC framework can be realized efficiently in practice, while remaining faithful to the diffusion-based interpretation developed above. A central advantage is that all stages of the method (diffusion-time selection, embedding, and clustering) can be implemented using repeated applications of the P-RW operator, without explicitly computing the iterated matrix $\mathbf{P}_{(\nu)}^t$ or its eigendecomposition.

***Low-dimensional approximations of $\mathbf{P}_{(\nu)}^t$.*** Rather than computing the iterated operator $\mathbf{P}_{(\nu)}^t$ directly via matrix exponentiation, its action on vectors or low-dimensional matrices can be computed through repeated applications of $\mathbf{P}_{(\nu)}$. For $\mathbf{Z}^{(0)} \in \mathbb{R}^{N \times d}$ randomly initialized (e.g. with uniform in $[0,1]$ or Gaussian entries), $\mathbf{Z}^{(\tau)}$ is computed as:

$$\mathbf{Z}^{(\tau)} = \mathbf{P}_{(\nu)}\mathbf{Z}^{(\tau-1)}, \quad \tau = 1, 2, ..., t. \tag{8}$$

After $t$ iterations, $\mathbf{Z}^{(t)}$ provides a low-dimensional representation of the action of $\mathbf{P}_{(\nu)}^t$ on the initial random matrix $\mathbf{Z}^{(0)}$, and can be seen as a random projection of $\mathbf{P}_{(\nu)}^t$ onto a $d$-dimensional space. This approach does not require computing the full $\mathbf{P}_{(\nu)}^t$ matrix, which reduces memory requirements and computational cost, especially when $d \ll N$. Moreover, it can be used for approximating the action of *any* iterated random walk operator, hence it can be beneficial for any PIC variant. These types of random projection techniques have been widely studied as Johnson–Lindenstrauss embeddings, see e.g. Freksen (2021). By the Johnson-Lindenstrauss property, for the low-dimensional embedding it holds with high probability that:

$$\left\|\mathbf{Z}^{(t)}(i,\cdot) - \mathbf{Z}^{(t)}(j,\cdot)\right\| \approx \left\|\mathbf{P}_{(\nu)}^t(i,\cdot) - \mathbf{P}_{(\nu)}^t(j,\cdot)\right\|.$$

Lin & Cohen (2010) used a similar approach in the undirected setting to compute low-dimensional approximations of the diffusion maps, considering $\mathbf{Z}$ to be a *single* vector instead of the matrix proposed here. As shown in Appendix D.3, there appears a performance plateau for $d \geq \sqrt{N}$ across all tested datasets, confirming that moderate dimensions do suffice. The exact choice of $d$ does not significantly impact clustering performance as long as it is reasonably large ($d = 1$ is not recommended).

***Diffusion time selection and entropy approximation.*** The operator entropy $\mathcal{H}(t)$ (Eq. 7) is defined solely over row entries of $\mathbf{P}_{(\nu)}^t$. This enables estimation of $\mathcal{H}(t)$ without forming the full matrix $\mathbf{P}_{(\nu)}^t$. Indeed, the $i$-th row $\mathbf{P}_{(\nu)}^t(i,\cdot)$ can be computed by iteratively right-multiplying the Kronecker delta vector $\delta_i$ by $\mathbf{P}_{(\nu)}$:

$$\mathbf{P}_{(\nu)}^t(i,\cdot) = (\delta_i^\mathsf{T}\mathbf{P}_{(\nu)})\,\mathbf{P}_{(\nu)}^{t-1}. \tag{9}$$

This allows the approximation of $\mathcal{H}(t)$ using only a sampled subset of vertices $\{i_1, i_2, ..., i_n\}$, without computing the full matrix $\mathbf{P}_{(\nu)}^t$, and therefore helps in reducing the cost of diffusion time selection:

$$\widehat{\mathcal{H}}(t) = \frac{N}{n}\sum_{r=1}^{n}\mathcal{H}_{i_r}(t), \tag{10}$$

where each $\mathcal{H}_{i_r}(t)$ is computed using $\mathbf{P}_{(\nu)}^t(i_r,\cdot)$ and $N/n$ is the reciprocal sampling ratio serving as a scaling factor.

**Proposition 3.6.** *The sampling-based estimator $\hat{\mathcal{H}}$ of the row-wise operator entropy $\mathcal{H}$ satisfies:*

- $\hat{\mathcal{H}}$ *is an unbiased estimator of $\mathcal{H}$.*
- *Its variance is upper-bounded:*

$$\mathrm{Var}(\hat{\mathcal{H}}(t)) \leq \frac{N^2}{n}\frac{N-n}{(N-1)}\frac{C(\pi_{(\nu)})^2}{4}.$$

- *For any $\eta > 0$, with probability at least $1 - \eta$:*

$$|\hat{\mathcal{H}}(t) - \mathcal{H}(t)| \leq \frac{N}{\sqrt{n}}\frac{C(\pi_{(\nu)})}{2}\sqrt{\frac{N-n}{(N-1)\eta}}.$$

The proof is deferred to Appendix E. The second point of Proposition 3.6 establishes that the estimator's variance decreases as the number of probes $n$ increases, with an upper bound depending on both $n$ and the graph size $N$. Practically, this means that a moderate number of probes (such as $n = \sqrt{N}$) achieves a favorable trade-off between estimation accuracy and computational cost: increasing $n$ reduces variance while keeping computations efficient, this aspect is also empirically shown in Appendix D.2, where it is seen that a moderate number of probes is sufficient to reliably identify the elbow of the curve $t \mapsto \mathcal{H}(t)$ across all tested datasets (see Fig. 8).

**Algorithm 1** Parametrized Power-Iteration Clustering

---

**Input:** $\mathbf{W} \in \mathbb{R}^{N \times N}$: adjacency matrix, $k$: number of clusters, $\nu$: vertex measure, $d$: dimensions for approximating the iterated random walk operator ($1 \leq d \leq N$, default: $d = \sqrt{N}$)
**Output:** $V$: graph $k$-partition

---

1: Compute the P-RW operator $\mathbf{P}_{(\nu)}$ (Eq. 3)
2: Select the right diffusion time $t$       (Secs. 3.4 and 3.5)
3: **if** $d < N$ **then**                         (Sec. 3.5)
4:       Compute $\mathbf{Z}_{(\nu)}^{(t)}$ approximating $\mathbf{P}_{(\nu)}^{t}$ in a $d$-dim. space
5: **else** Compute $\mathbf{P}_{(\nu)}^{t}$ by iterating $t$ times the operator $\mathbf{P}_{(\nu)}$
6: **end if**
7: Apply $k$-means on the rows of $\mathbf{Z}_{(\nu)}^{(t)}$ (or $\mathbf{P}_{(\nu)}^{t}$ if $d = N$) to obtain the clustering $V$
8: **return** $V$

---

**Clustering.** The low-dimensional representation $\mathbf{Z}^{(t)}$ can be used to perform clustering, treating each row as a $d$-dimensional embedding of a vertex. This approach leverages the diffusion dynamics captured in $\mathbf{Z}^{(t)}$ to group vertices based on their connectivity patterns in the graph. Same as in the spectral clustering context, the idea is that the new data representation (here the rows of $\mathbf{P}_{(\nu)}^{t}$ or its approximation $\mathbf{Z}^{(t)}$) reveal the cluster structure of the data, so that a simple method such as $k$-means suffices to find the final clusters.

## 4. Experiments

### 4.1. Setup

**Baselines.** ParPIC is compared against a comprehensive suite of state-of-the-art digraph clustering algorithms. The selection covers the main theoretical approaches (Hermitian vs. Random Walk) and computational paradigms (Spectral vs. Power-Iteration) in the literature; additional details are given in Appendix B.2. The baselines used are summarized in Tab. 1, and are categorized into three distinct families:

- *Hermitian spectral methods:* Approaches based on the eigen-decomposition of Hermitian matrices (Simple-Herm, Herm-SC, Herm-RW). While these recover real spectra, they generally lack a diffusion interpretation.
- *Random-walk spectral methods:* Methods relying on the spectrum of transition matrices or their Laplacians (DSC+, DD-Sym, PR-SC, GSC). These are grounded in diffusion geometry, but require expensive eigen-decompositions.
- *Power-iteration methods:* Scalable alternatives that approximate diffusion embeddings via matrix multiplications. They are based on the natural random walk of the directed graph (PIC), the symmetrized random walk (S-PIC, computed via the normalization of $\mathbf{A} + \mathbf{A}^{\mathsf{T}}$), or the teleportation-based random walk (PR-PIC). ParPIC is an instance of this family. For all methods of this family, the time selection criterion and the low-dimensional ap-

*Table 1.* **Compared methods: abbreviations and references.** Methods are grouped into spectral methods (top), which require the eigen-decomposition of various digraph operators, and power-iteration clustering methods (bottom), which build vertex embeddings via iterative application of a diffusion operator. Spectral methods are generally more computationally expensive due to the eigen-decomposition step, while power-iteration methods can be more scalable.

| Abbreviation | Reference | Relies on eigen-decomp. | Diffusion interpretation |
|---|---|---|---|
| Simple-Herm | Laenen & Sun (2020) | ✓ | ✗ |
| Herm-SC | Cucuringu et al. (2020) | ✓ | ✗ |
| Herm-RW | Mohar (2020) | ✓ | ✗ |
| DD-Sym | Satuluri & Parthasarathy (2011) | ✓ | ✗ |
| DSC+ | Chung (2005) | ✓ | ✗ |
| GSC | Sevi et al. (2025) | ✓ | ✓ |
| Sym-SC | Satuluri & Parthasarathy (2011) | ✓ | ✓ |
| PIC | Lin & Cohen (2010) | ✗ | ✓ |
| PR-PIC | Lin & Cohen (2010) | ✗ | ✓ |
| S-PIC | Lin & Cohen (2010) | ✗ | ✓ |
| ParPIC | This work | ✗ | ✓ |

proximations of the iterated operators derived in Sec. 3.5 are used. This allows to mostly focus on testing the modeling capacity of the proposed P-RW operator.

**Datasets.** We report results on a diverse collection of digraphs (details are in Appendix B.1), categorized into two types:

- *K-NN digraphs:* Unweighted digraphs constructed from vector data by connecting each vertex to its $K$-nearest neighbors. Ten UCI datasets are used: Iris, Wine, Glass, WDBC, Control Chart, Segmentation, Seeds, Olivetti, Vertebral, and Yeast.
- *Directed networks:* Graphs with intrinsic directionality, namely various synthetic ones from the Directed Stochastic Block Model (DiSBM), and the real-world Political Blogs (PolBlogs) and Email networks.

Note that both Political Blogs and Email networks contain a significant amount of sinks and sources and are also nonstrongly connected, making them challenging for diffusion- or spectral-based methods.

**Evaluation metrics.** Clustering performance is evaluated using Adjusted Mutual Information (AMI), which quantifies agreement between predicted clusters and ground truth while accounting for chance:

$$\text{AMI}(U, V) = \frac{\text{MI}(U, V) - \mathbb{E}[\text{MI}(U, V)]}{\max(H(U), H(V)) - \mathbb{E}[\text{MI}(U, V)]} \in [0, 1],$$

where $U$ and $V$ are two clusterings, MI is the mutual information, and $H$ is the entropy. Higher AMI values indicate better clustering performance. The *Performance Relative to the Best* (PRB) measure is also reported, defined as:

$$\text{PRB}(l) = \frac{1}{M} \sum_{m=1}^{M} \frac{\text{AMI}_{l,m}}{\max_k \text{AMI}_{k,m}} \in [0, 1],$$

*Table 2.* **Clustering results (AMI) on $K$-NN digraphs.** Best results are in **bold**, second best are underlined, stds appear in parentheses.

| Methods | PRB | Datasets | | | | | | | | | |
|---|---|---|---|---|---|---|---|---|---|---|---|
| | | Iris | Wine | Glass | WDBC | Control Chart | Segmentation | Seeds | Olivetti | Vertebral | Yeast |
| Herm-SC | 0.40 | 0.18 (0.04) | 0.47 (0.06) | 0.16 (0.02) | 0.15 (0.09) | 0.45 (0.05) | 0.04 (0.01) | 0.29 (0.04) | 0.51 (0.02) | 0.13 (0.02) | 0.11 (0.01) |
| Herm-RW | 0.40 | 0.20 (0.03) | 0.49 (0.06) | 0.16 (0.02) | 0.13 (0.13) | 0.45 (0.03) | 0.06 (0.01) | 0.26 (0.03) | 0.53 (0.02) | 0.11 (0.06) | 0.13 (0.01) |
| Simple-Herm | 0.56 | 0.23 (0.01) | 0.81 (0.00) | 0.19 (0.02) | 0.08 (0.02) | 0.64 (0.02) | 0.08 (0.01) | 0.52 (0.04) | 0.61 (0.01) | 0.22 (0.04) | 0.15 (0.02) |
| DD-Sym | 0.39 | 0.26 (0.03) | 0.34 (0.06) | 0.22 (0.02) | 0.13 (0.06) | 0.23 (0.03) | 0.04 (0.00) | 0.27 (0.05) | 0.45 (0.01) | 0.12 (0.06) | 0.16 (0.02) |
| DSC+ | 0.59 | 0.43 (0.06) | 0.54 (0.07) | 0.21 (0.02) | 0.26 (0.05) | 0.59 (0.05) | 0.07 (0.01) | 0.58 (0.15) | 0.60 (0.01) | 0.20 (0.10) | 0.19 (0.02) |
| GSC | 0.73 | 0.45 (0.07) | 0.48 (0.12) | 0.24 (0.02) | 0.64 (0.00) | 0.48 (0.04) | 0.12 (0.02) | 0.66 (0.08) | 0.69 (0.01) | 0.42 (0.10) | 0.23 (0.01) |
| Sym-SC | 0.85 | 0.58 (0.11) | 0.61 (0.11) | 0.26 (0.02) | **0.70 (0.00)** | 0.57 (0.04) | 0.19 (0.03) | **0.76 (0.06)** | **0.71 (0.01)** | **0.51 (0.01)** | 0.28 (0.01) |
| PIC | 0.86 | 0.69 (0.12) | **0.86 (0.03)** | 0.25 (0.03) | 0.66 (0.06) | 0.67 (0.06) | 0.22 (0.05) | 0.63 (0.09) | 0.60 (0.02) | 0.46 (0.04) | 0.28 (0.01) |
| PR-PIC | 0.84 | 0.63 (0.14) | **0.86 (0.04)** | 0.25 (0.03) | 0.64 (0.07) | 0.66 (0.05) | 0.21 (0.05) | 0.61 (0.09) | 0.60 (0.01) | 0.45 (0.05) | 0.28 (0.01) |
| S-PIC | **0.96** | **0.77 (0.05)** | 0.85 (0.01) | **0.28 (0.03)** | 0.67 (0.03) | **0.74 (0.04)** | **0.49 (0.07)** | 0.73 (0.04) | 0.65 (0.01) | 0.44 (0.05) | **0.29 (0.00)** |
| ParPIC | **0.96** | 0.76 (0.07) | 0.85 (0.03) | 0.26 (0.03) | 0.69 (0.02) | 0.73 (0.04) | 0.48 (0.05) | 0.70 (0.07) | 0.66 (0.01) | 0.47 (0.06) | 0.29 (0.01) |

*Table 3.* **Clustering results (AMI) on directed networks.** The tested digraphs include various synthetic directed stochastic block models (DiSBM) and real-world networks (PolBlogs, Email-Eu). Best results are in **bold**, second best are underlined, stds appear in parentheses.

| Methods | PRB | Datasets | | | | |
|---|---|---|---|---|---|---|
| | | DiSBM-Baseline | DiSBM-Chain | DiSBM-CP | PolBlogs | Email-Eu |
| Herm-SC | 0.34 | 0.00 (0.00) | 0.47 (0.07) | 0.22 (0.21) | 0.13 (0.01) | 0.32 (0.01) |
| Herm-RW | 0.40 | 0.00 (0.00) | 0.44 (0.05) | 0.28 (0.23) | 0.21 (0.01) | 0.35 (0.01) |
| Simple-Herm | 0.50 | 0.97 (0.00) | 0.42 (0.10) | 0.19 (0.10) | 0.00 (0.00) | 0.44 (0.01) |
| DD-Sym | 0.72 | **1.00 (0.00)** | 0.85 (0.16) | 0.56 (0.14) | 0.08 (0.01) | 0.44 (0.01) |
| DSC+ | 0.33 | **1.00 (0.00)** | 0.14 (0.03) | 0.49 (0.13) | 0.00 (0.00) | 0.00 (0.00) |
| GSC | 0.50 | **1.00 (0.00)** | 0.14 (0.03) | 0.66 (0.08) | 0.02 (0.02) | 0.32 (0.05) |
| Sym-SC | 0.57 | **1.00 (0.00)** | 0.57 (0.02) | 0.65 (0.10) | 0.02 (0.01) | 0.26 (0.05) |
| PIC | 0.51 | 0.99 (0.04) | 0.28 (0.20) | 0.96 (0.00) | 0.10 (0.16) | 0.01 (0.04) |
| PR-PIC | 0.55 | 0.99 (0.00) | 0.25 (0.20) | 0.96 (0.05) | 0.16 (0.17) | 0.05 (0.06) |
| S-PIC | 0.65 | **1.00 (0.00)** | 0.57 (0.01) | 0.59 (0.01) | 0.01 (0.00) | **0.49 (0.01)** |
| ParPIC | **1.00** | **1.00 (0.00)** | **0.93 (0.02)** | **1.00 (0.00)** | **0.39 (0.14)** | 0.48 (0.02) |

where $M$ is the number of datasets, $\text{AMI}_{k,m}$ is the mean AMI of method $k$ on dataset $m$. PRB normalizes performance across datasets, with values closer to $1$ indicating consistently strong performance.

### 4.2. Results

We evaluate ParPIC using 10 baseline methods across two distinct experimental settings: $K$-NN digraphs constructed from real-world point-cloud data (Tab. 2), and intrinsically directed networks including synthetic DiSBMs and real digraphs (Tab. 3). Tabular results report average AMI and standard deviation (std) over $100$ independent runs, with associated PRB scores.

***Results on K-NN digraphs (Tab. 2).*** The digraphs induced by $K$-NN construction exhibit high reciprocity and homogeneous degree distributions (see Tab. 5). In this context, the proposed method performs better than all other methods except our S-PIC implementation, both having a PRB of 0.96. In this regime where edge directionality has limited influence on the induced diffusion geometry, methods that use symmetrized operators or approximations based on undirected operators are also effective. Consequently, explicitly modeling directionality yields only marginal gains. Nevertheless, the three groups obtain different performance ranges, with the Power-Iteration group obtaining the highest scores, while the Hermitian-based group shows generally lower PRB (0.40-0.56) compared to the other two method families, which suggests that their clustering objective is misaligned with the problem at hand. Importantly, the proposed approach does not suffer from a performance loss in this setting, while retaining the computational benefits of an eigen-free formulation.

***Results on intrinsically directed graphs (Tab. 3).*** On graphs with pronounced directional asymmetries, heterogeneous degrees, and low reciprocity, the proposed method consistently outperforms competing approaches, including symmetrization-based, teleportation-based, Hermitian spectral, and existing power-iteration methods. Performance improvements are most evident in settings where clusters are defined by asymmetric flow patterns, such as source-sink or core-periphery structures. In these cases, symmetrization obscures directional information, while teleportation alters the underlying dynamics. By contrast, the proposed P-RW preserves directionality while ensuring reversibility, yielding diffusion embeddings that more accurately reflect the latent cluster structure. Aggregated across datasets, the proposed method achieves the strongest normalized performance-relative scores (PRB of 1.00), indicating robust behavior across diverse graphs.

These results highlight a clear distinction between weakly and strongly directed settings: while many methods perform similarly in the former, explicitly incorporating directionality into the diffusion process is critical in the latter.

The spectral-based methods relying on eigen-decomposition generally underperform compared to power-iteration approaches, with the exception of DD-Sym, likely due to sensitivity to graph irregularities or/and sensitivity of the spectra to symmetrization techniques. This underpins the importance of symmetrization techniques, as different symmetrizations lead to very different results. Overall, the proposed ParPIC approach provides a unified, scalable solution for clustering directed graphs without eigen-decomposition.

### 4.3. Experiments on Degree Heterogeneity

We further demonstrate the advantages of our P-RW operator in handling digraphs with cluster-level degree heterogeneity, a scenario where symmetrization-based methods often struggle. Fig. 2 shows the impact of such heterogeneity on clustering performance; we specifically analyze the 3-cluster Core-Periphery (C-P) DiSBM model:

$$\mathbf{Q}_\rho = \begin{bmatrix} 0.05 & \rho & \rho \\ 0.01 & 0.05 & 0.01 \\ 0.01 & 0.01 & 0.05 \end{bmatrix}, \qquad (11)$$

where the parameter $\rho$ controls cluster-level degree heterogeneity: higher $\rho$ values induce more pronounced differences in out-degrees between clusters. To compare ParPIC, S-PIC, PIC and DD-Sym, we generate graphs with $3 \times 1300$ vertices, and vary $\rho \in [0.1, ..., 0.4]$. Fig. 2a shows that symmetrization-based methods degrade significantly when $\rho$ increases, while ParPIC remains robust; the operators for the $\rho$ values just before and just after the abrupt degradation of S-PIC are shown in Fig. 2b. This suggests that our P-RW operator handles cluster-level degree heterogeneity effectively by adapting to the directed structure without being compromised by asymmetric vertex degrees. Additional experiments on DiSBMs are explored in Appendix C, including the impact of the number of clusters, impact of the flow-strength on the chain DiSBM, and joint analysis on the size of the 'sender' cluster and its strength in the core-periphery DiSBM.

Scalability results (Fig. 3) show that ParPIC outperforms spectral clustering in run-time, even with full projection, and scales very efficiently with random projections.

### 4.4. Summary and Discussion

The experimental results demonstrate that ParPIC effectively balances computational efficiency with clustering accuracy across a range of directed graph structures. In $K$-NN digraphs with high reciprocity, it matches the performance of leading methods while avoiding costly eigen-decompositions. In intrinsically directed networks with pronounced asymmetries, it outperforms all baselines by preserving directional information in the diffusion process. The proposed vertex measure design and diffusion time selection strategy are key contributors to this success, enabling

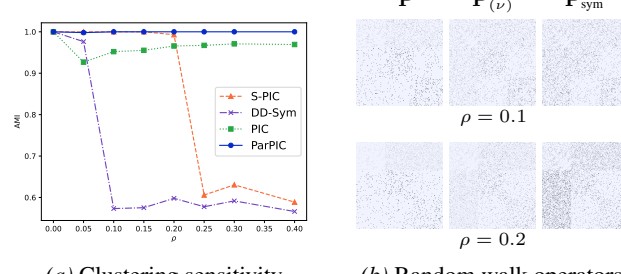

*(a)* Clustering sensitivity     *(b)* Random walk operators

*Figure 2.* **Sensitivity of different methods to cluster-level out-degree heterogeneity.** (a) Average performance on 50 runs, while varying $\rho$ value in the DiSBM model of Eq. 11. (b) PIC, S-PIC and ParPIC operators at different $\rho$ values.

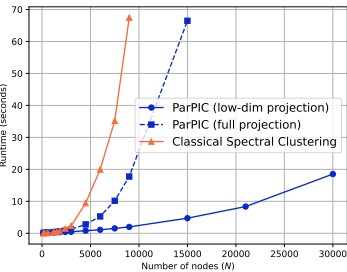

*Figure 3.* **Scaling of runtime with graph size (3-cluster DiSBM-CP).** ParPIC, with the default approximation of the iterated P-RW operator ($\mathbf{P}_{(\nu)}^t$), compared to ParPIC with the full computation of the P-RW operator (any typical PIC variant shares this complexity) and the classical spectral clustering. The proposed method demonstrates significantly better scalability.

flexible adaptation to diverse graph topologies. Our analysis (Appendices D.3 and D.4) validates these design choices: the default $\gamma = 0.5$ performs well across most settings, the embedding dimension $d \simeq \sqrt{N}$ suffices for stable performance, the entropy-based time selection consistently identifies meaningful diffusion scales, and the sampling-based variant recovers the same elbows using $\sqrt{N}$ probes.

To focus on the comparison of the modeling capacity of different random walk operators to the introduced P-RW operator, all PIC variants use our entropy-based time selection and low-dimensional approximations (Secs. 3.4 and 3.5). This controls for implementation improvements, making the reported gaps a conservative estimate, since our methodological enhancements (time selection, random projection) are credited to the baselines as well.

Our findings underscore the importance of explicit modeling of edge directionality when clustering directed graphs, and support the design choices behind the ParPIC framework. The results highlight the limitations of symmetrization and teleportation-based approaches in capturing the true community structure of directed graphs, emphasizing the need for methods that respect the inherent directionality of edges. When directionality plays a 'high-level' role in defining clus-

ters, e.g. in cases where clusters are defined by asymmetric flow patterns, the advantages of ParPIC become particularly pronounced. This is underpinned by the superior performance observed in the DiSBM core-periphery and chain structures, where traditional methods fail due to their inability to adequately capture the directional dynamics that are crucial for accurate clustering.

## 5. Conclusion

In this work, we have introduced Parametrized Power-Iteration Clustering (ParPIC), a novel approach for clustering directed graphs based on a parametrized random walk (P-RW) operator. By designing a flexible vertex measure that captures the edge directionality and the random walk dynamics, the P-RW operator effectively balances in-degree and out-degree information, allowing for improved clustering performance across degree-heterogeneous digraphs. We also proposed an efficient strategy for diffusion time selection that identifies the elbow in the entropy curve of the iterated operator, enhancing the adaptability and performance of diffusion-based clustering on different graph topologies. Clustering experiments on both synthetic and real-world digraphs demonstrate that ParPIC outperforms spectral techniques revolving around symmetrization of the adjacency matrix, and is competitive with power-iteration methods, where we observe gains in performance when edge directionality is crucial for the random walk dynamics. Future work includes extensions to dynamic digraphs, semi-supervised learning, and alternative vertex measure designs that incorporate additional vertex attributes or edge weights.

## Acknowledgments

We would like to thank Gaëtan Serré and Malik Hacini for the insightful discussions. Harry Sevi, Gwendal Debaussart-Joniec, and Argyris Kalogeratos acknowledge the support of the Industrial Analytics and Machine Learning (IdAML) Chair hosted at ENS Paris-Saclay, Université Paris-Saclay. Matthieu Jonckheere was funded by the International Centre for Mathematics and Computer Science (CIMI) in Toulouse.

## Software and Data

Code for the proposed method and experiments is available at: https://github.com/Gwendal-Debaussart/parpic.

## Impact Statement

This paper presents work whose goal is to advance the field of Machine Learning. There are many potential societal consequences of our work, none of which we feel must be specifically highlighted here.

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

# A. Notation Table

The main notation used throughout the paper are summarized in Tab. 4.

*Table 4.* Notation.

| Symbol | Description |
|---|---|
| $\mathbf{A}_{ij}, \mathbf{A}(i,j)$ | Entry $(i,j)$ of the matrix $\mathbf{A}$ |
| $f_i, f(i)$ | Entry $i$ of the vector $f$, evaluation of the vertex-level function $f$ at vertex $i$ |
| $\mathbb{1}\{A\}$ | Indicator function of the set $A$ |
| $\delta_i$ | The Kronecker delta (one hot) vector, vector with 1 at position $i$ and 0 elsewhere |
| $\mathcal{G} = (V, E, w)$ | Directed graph with vertex set $V$, edge set $E$, and weight function $w$ |
| $\mathbf{W}$ | Adjacency matrix of a (di-)graph |
| $N$ | Number of vertices in the graph |
| $d_{\text{out}}(i), \mathbf{D}_{\text{out}}$ | Out-degree of vertex $i$, out-degree matrix |
| $d_{\text{in}}(i), \mathbf{D}_{\text{in}}$ | In-degree of vertex $i$, in-degree matrix |
| $\mathbf{P}$ | Random walk transition matrix (Sec. 2) |
| $\mathcal{D}_t^2$ | Diffusion distance (Sec. 2) |
| $\nu, \xi$ | Vertex measure and its push-forward by the random walk operator $\mathbf{P}$ ($\xi = \mathbf{P}^\mathsf{T}\nu$, Sec. 3) |
| $\mathbf{D}_\nu$ | Diagonal matrix with vertex measure $\nu$ on its diagonal |
| $\mathbf{P}_{(\nu)}$ | Parametrized random walk (P-RW) operator |
| $\mathcal{D}_{t,(\nu)}^2$ | Parametrized diffusion distance (Sec. 3) |

# B. Experimental details

### B.1. Datasets

A summary of the datasets used in the experiments is provided in Tab. 5, including the number of vertices, edges, clusters, and some statistics. Reciprocity is defined as the ratio of the number of bidirectional edges to the total number of edges in the directed graph, providing a measure of how many connections are mutual. Gini coefficients for in-degrees and out-degrees quantify the inequality in the distribution of connections among vertices, with higher values indicating greater disparity. Cluster-level reciprocity (C-L Reciprocity) measures the balance of inter-cluster connections, indicating how reciprocal the connections are between different clusters. They are computed as:

$$\text{Reciprocity}(\mathbf{W}) = \sum_{i,j} \frac{\mathbf{W}_{ij} \cdot \mathbf{W}_{ji}}{\sum_{i,j} \mathbf{W}_{ij}}, \tag{12}$$

$$\text{Gini}(x) = \frac{\sum_{i=1}^N \sum_{j=1}^N |x_i - x_j|}{2N \sum_{i=1}^N x_i}, \tag{13}$$

$$\text{C-L Reciprocity}(\mathbf{W}, y) = \frac{1}{k(k-1)} \sum_{a \neq b} \frac{2 \cdot \min\left(E(a,b), E(b,a)\right)}{E(a,b) + E(b,a)}, \tag{14}$$

where $E(a,b) = \sum_{i \in a, j \in b} \mathbf{W}_{ij}$ is the number of edges from cluster $a$ to cluster $b$, $y$ is a cluster assignment, and $k$ is the total number of clusters.

*K-NN digraphs.* Digraphs are constructed from vector data by connecting each vertex to its $K$-nearest neighbors based on Euclidean distance. Ten standard datasets from the UCI repository are used: Iris, Wine, Glass, WDBC, Control Chart, Segmentation, Seeds, Olivetti, Vertebral, and Yeast. These datasets vary in size, dimensionality, and class distribution, providing a diverse set of benchmarks for evaluating clustering algorithms on $K$-NN digraphs.

*Synthetic directed stochastic block models (DiSBM).* Synthetic digraphs are generated using the directed Stochastic Block Model (DiSBM) framework. In this model, vertices are partitioned into $k$ clusters, and the probability of a directed edge from vertex $i$ to vertex $j$ depends on the clusters to which these vertices belong. Various configurations of DiSBM are considered. The specific parameters used for generating the DiSBM graphs in the experiments are as follows, the $\mathbf{Q}$ matrix represents the inter-cluster connection probabilities, and $m$ is the number of vertices per cluster:

- **DiSBM-CP:** This configuration creates a core-periphery structure, where one cluster (the core) has high connectivity to

*Table 5.* **Dataset statistics.** Summary of the datasets used in the experiments, including the number of vertices, edges, and clusters and some statistics. The first group contains $K$-NN based digraphs, while the second one contains 'natural' digraphs. Gini coefficients for in-degrees and out-degrees quantify the inequality in the distribution of connections among vertices, with higher values indicating greater disparity. C-L Reciprocity refers to the cluster-level reciprocity (Eq. 14). By definition, the $K$-NN digraphs have uniform out-degrees and thus Gini($d_{\text{out}}$) is not applicable (—).

| Dataset | Vertices | Edges | Clusters | Reciprocity | Gini ($d_{\text{out}}$) | Gini ($d_{\text{in}}$) | C-L Reciprocity |
|---|---|---|---|---|---|---|---|
| Iris | 150 | 450 | 3 | 0.64 | — | 0.32 | 0.30 |
| Wine | 178 | 534 | 3 | 0.58 | — | 0.37 | 0.20 |
| Glass | 214 | 642 | 6 | 0.57 | — | 0.33 | 0.43 |
| WDBC | 569 | 1707 | 2 | 0.48 | — | 0.38 | 0.77 |
| Control Chart | 600 | 1800 | 6 | 0.65 | — | 0.34 | 0.02 |
| Segmentation | 2310 | 6930 | 7 | 0.72 | — | 0.23 | 0.27 |
| Seeds | 210 | 630 | 3 | 0.69 | — | 0.26 | 0.60 |
| Olivetti | 400 | 1200 | 40 | 0.67 | — | 0.32 | 0.05 |
| Vertebral | 310 | 930 | 3 | 0.57 | — | 0.34 | 0.92 |
| Yeast | 1484 | 4452 | 10 | 0.58 | — | 0.30 | 0.79 |
| DiSBM-CP | 4000 | 2.5m | 3 | 0.04 | 0.54 | 0.29 | 0.35 |
| DiSBM-Chain | 1500 | 340k | 3 | 0.02 | 0.30 | 0.30 | 0.02 |
| DiSBM-Baseline | 1500 | 52k | 3 | 0.04 | 0.09 | 0.09 | 0.98 |
| PolBlogs | 1222 | 16k | 2 | 0.24 | 0.70 | 0.80 | 0.93 |
| Email-Eu | 1005 | 25k | 42 | 0.72 | 0.61 | 0.54 | 0.54 |

the other two clusters (the periphery), while the periphery clusters have low connectivity among themselves. This results in highly asymmetric flow patterns, which are challenging for clustering algorithms that do not account for directionality. Moreover, the degree distribution is heterogeneous, with the core cluster having significantly higher degrees than the periphery clusters. The parameters are set as:

$$k = 3 \quad \text{clusters}, \quad m = [1300, 1300, 1300] \text{ vertices per cluster, and } \mathbf{Q} = \begin{bmatrix} 0.05 & 0.6 & 0.6 \\ 0.02 & 0.05 & 0.02 \\ 0.02 & 0.02 & 0.05 \end{bmatrix}.$$

- **DiSBM-Chain:** This configuration creates a chain-like structure, where each cluster primarily connects to the next cluster in sequence. This results in a directed flow of connections from the first cluster to the last, with minimal backward connections. The degree distribution is relatively uniform across clusters, but the directional flow creates challenges for clustering algorithms that do not consider edge directionality. The parameters are set as:

$$k = 3 \quad \text{clusters}, \quad m = [500, 500, 500] \text{ vertices per cluster, and } \mathbf{Q} = \begin{bmatrix} 0.05 & 0.6 & 0.0 \\ 0.01 & 0.05 & 0.6 \\ 0.0 & 0.01 & 0.05 \end{bmatrix}. \tag{15}$$

- **DiSBM-Baseline:** This configuration creates a balanced structure, where each cluster has similar intra-cluster and inter-cluster connection probabilities. The resulting graph has a more uniform degree distribution and less pronounced directional flow patterns, making it a baseline scenario for evaluating clustering algorithms on directed graphs. Moreover, even though the graph is directed, the general structure isn't, making it more suitable for symmetrization-based methods. The parameters are set as:

$$k = 3 \quad \text{clusters}, \quad m = [500, 500, 500] \text{ vertices per cluster, and } \mathbf{Q} = \begin{bmatrix} 0.05 & 0.01 & 0.01 \\ 0.01 & 0.05 & 0.01 \\ 0.01 & 0.01 & 0.05 \end{bmatrix}.$$

### B.2. Method Implementations and Hyperparameters

The implementations and specific parameter settings for all compared methods are detailed below. For all spectral methods, the eigenvectors corresponding to the smallest (or largest, depending on the operator) eigenvalues are computed and $k$-means is applied to the embedding. All power-iteration methods use a time parameter $t$ selected via the entropy strategy described in Sec. 3.4. They are based on the random projection technique detailed in Sec. 3.5, with dimension set to $d = \sqrt{N}$ in all cases.

***Hermitian-based spectral clustering methods.*** For the Hermitian-based methods, two different configurations are considered, the *Hermitian* and *RW-Hermitian* (Cucuringu et al., 2020; Mohar, 2020), and the *Simple-Herm* (Laenen & Sun, 2020).

Hermitian methods construct complex-valued Hermitian matrices to encode the directionality of edges in the phase of the spectrum (Guo & Mohar, 2017). To do so, two main approaches exist, either using the imaginary unit $i$ to encode directionality, or complex roots of unity. In particular, the *Hermitian* and *RW-Hermitian* methods are based on the $\mathbf{H}$ matrix, while the *Simple-Herm* method is based on the $\mathbf{S}$ matrix. They are defined as:

$$\mathbf{H}_{ij} = \begin{cases} i & \text{if } \mathbf{W}_{ij} > 0 \text{ and } \mathbf{W}_{ji} = 0 \\ -i & \text{if } \mathbf{W}_{ij} = 0 \text{ and } \mathbf{W}_{ji} > 0 \\ 1 & \text{if } \mathbf{W}_{ij} > 0 \text{ and } \mathbf{W}_{ji} > 0 \\ 0 & \text{otherwise,} \end{cases} \quad \text{and} \quad \mathbf{S}_{ij} = \begin{cases} \mathbf{W}_{ij} \cdot \omega & \text{if } \mathbf{W}_{ij} > 0 \text{ and } \mathbf{W}_{ji} = 0 \\ \mathbf{W}_{ij} \cdot \bar{\omega} & \text{if } \mathbf{W}_{ij} = 0 \text{ and } \mathbf{W}_{ji} > 0 \\ 1 & \text{if } \mathbf{W}_{ij} > 0 \text{ and } \mathbf{W}_{ji} > 0 \\ 0 & \text{otherwise,} \end{cases}$$

where $i$ is the imaginary unit, $\omega$ is a primitive $k$-th root of unity (i.e. $\omega = \exp(2\pi i/k)$, for some integer $k > 0$), and $\mathbf{W}$ is the adjacency matrix. *Hermitian* spectral clustering either use $\mathbf{H}$ directly (Herm-SC) or the 'random-walk' normalized version $\mathbf{H}_{RW}$, normalized according to $\sum_j |\mathbf{H}_{ij}|$. The Simple-Herm method uses the matrix $\mathbf{L}_{\text{SH}} = \mathbf{I} - (\mathbf{D}_{\text{out}} + \mathbf{D}_{\text{in}})^{1/2}\mathbf{S}(\mathbf{D}_{\text{out}} + \mathbf{D}_{\text{in}})^{-1/2}$. In both cases, matrices are Hermitian and thus admit a real spectrum, which is used for traditional spectral clustering. Hermitian matrices are by construction complex-valued, leading to the loss of interpretability as random walk operators. Moreover, these methods are based on flow-based clustering objectives that may not align well with the underlying community structures in the tested digraphs (see, e.g. Cucuringu et al. (2020)).

***Baselines based on symmetrization and directed Laplacians.*** This group of methods typically constructs a modified adjacency or Laplacian matrix that captures the directed nature of the graph while maintaining the real-valued entries, allowing for the application of standard spectral clustering techniques. *DD-Sym* uses the bibliographic symmetrization (Satuluri & Parthasarathy, 2011) $\mathbf{A}\mathbf{A}^\top + \mathbf{A}^\top\mathbf{A}$. *Sym-SC* (Satuluri & Parthasarathy, 2011) uses the symmetrized random walk operator $\mathbf{P}_{\text{sym}} \propto \mathbf{A} + \mathbf{A}^\top$. *DSC+* uses the directed Laplacians (Chung, 2005) defined as $\mathbf{L}_C = \mathbf{D}_\pi - (\mathbf{D}_\pi\mathbf{P} + \mathbf{P}^\top\mathbf{D}_\pi)/2$, where $\pi$ is the ergodic law of the natural random walk $\mathbf{P}$. When $\pi$ does not exist, $\mathbf{P}^t(i, \cdot)$ is used for a sufficiently large $t$. Generalized Spectral Clustering *GSC* uses the generalized Laplacian framework (Sevi et al., 2025), which defines a family of directed Laplacians based on vertex measures; in this context, the vertex measure defined in Sec. 3 is used.

***Power-Iteration Clustering baselines.*** Three baselines based on the power-iteration clustering (PIC) framework (Lin & Cohen, 2010) are considered. The original PIC method (PIC) uses the natural random walk operator $\mathbf{P}$. PageRank-PIC (PR-PIC) uses the PageRank transition matrix $\mathbf{P}_\alpha = \alpha\mathbf{P} + (1 - \alpha)\frac{1}{N}\mathbf{1}\mathbf{1}^\top$ with $\alpha = 0.85$. Symmetric-PIC (S-PIC) uses the symmetrized random walk operator $\mathbf{P}_{\text{sym}} \propto \mathbf{A} + \mathbf{A}^\top$. As stated in the introduction, every method uses the time selection scheme proposed in Sec. 3.4 and the random projection approach of Sec. 3.5.

## C. Additional Experiments

***Impact of the flow strength on DiSBM-Chain.*** Additional experiments on the chain-structured DiSBM (Eq. 16) are shown: the flow impact on clustering performance of different algorithms is analyzed, according to the following 3-cluster model:

$$k = 3 \quad \text{clusters,} \quad m = [500, 500, 500] \text{ vertices per cluster, and } \mathbf{Q}_\rho = \begin{bmatrix} 0.05 & \rho & 0.0 \\ 0.01 & 0.05 & \rho \\ 0.0 & 0.01 & 0.05 \end{bmatrix}. \quad (16)$$

Fig. 4 visualizes the natural random walk operator $\mathbf{P}$, the symmetrized operator $\mathbf{P}_{\text{sym}}$, and our parametrized random walk operator $\mathbf{P}_{(\nu)}$ on DiSBM-Chain($\rho$) (Eq. 15). The symmetrized operator $\mathbf{P}_{\text{sym}}$ introduces non-existing connections between clusters due to high out-degrees from cluster 1, which obscures the cluster boundaries and degrades clustering performance. In contrast, the P-RW operator $\mathbf{P}_{(\nu)}$ effectively balances in-degree and out-degree information, preserving the cluster structure when incorporating these weighted edges.

***Joint analysis of cluster size and flow strength DiSBM-CP.*** The joint impact of size imbalance and out-degree heterogeneity on clustering performance is examined for DiSBM-CP($\rho, m_1$). The following 3-cluster model (Eq. 17) is used:

$$k = 3 \quad \text{clusters,} \quad m = [m_1, 1300, 1300] \text{ vertices per cluster, and } \mathbf{Q}_\rho = \begin{bmatrix} 0.05 & \rho & \rho \\ 0.01 & 0.05 & 0.01 \\ 0.01 & 0.01 & 0.05 \end{bmatrix}. \quad (17)$$

The comparison focuses on ParPIC with the symmetrization-based power-iteration method (S-PIC). By varying the core cluster size $m_1$ and the parameter $\rho$, the size imbalance and out-degree heterogeneity of the core cluster are controlled,

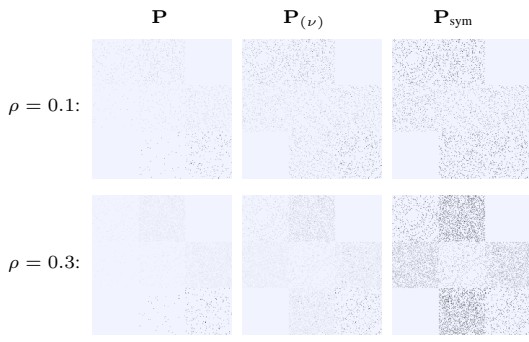

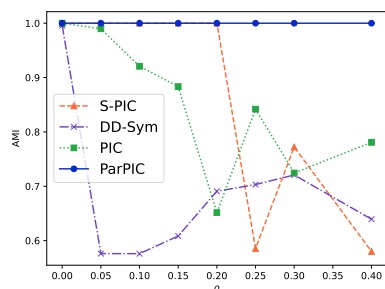

*(a)* Effect of symmetrization on random walk operators

*(b)* Clustering sensitivity to flow strength

*Figure 4.* **Experiments on DiSBM-Chain**($\rho$). (a) Natural ($\mathbf{P}$), parametrized ($\mathbf{P}_{(\nu)}$) and symmetrized ($\mathbf{P}_{\text{sym}}$) random walk operators on the DiSBM-Chain($\rho$) model (Eq. 16), according to different flow strengths. (b) Clustering sensitivity to flow strength; the proposed ParPIC remains stable across varying $\rho$ values, while variants of PIC significantly degrade as the flow increases.

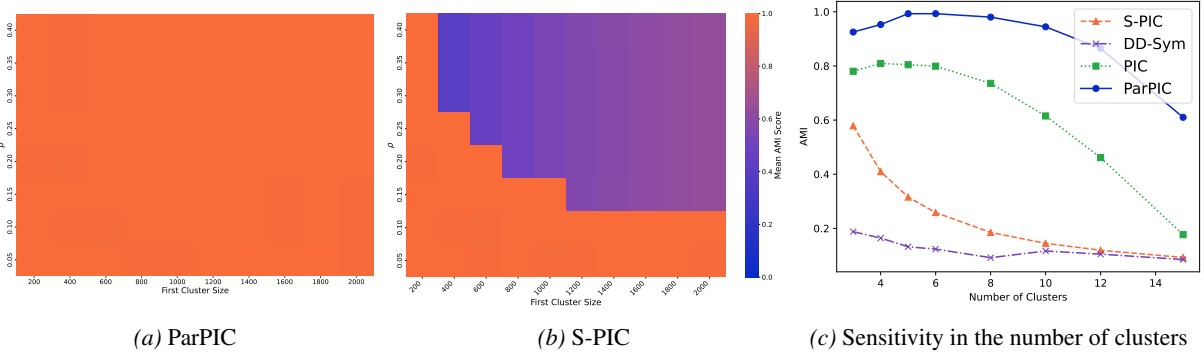

*(a)* ParPIC

*(b)* S-PIC

*(c)* Sensitivity in the number of clusters

*Figure 5.* **Additional experiments on DiSBM-CP.** (a)-(b) Clustering performance (AMI) of ParPIC and S-PIC on DiSBM-CP($\rho, m_1$) (Eq. 17). (c) Clustering performance (AMI) of different methods on DiSBM-CP($k$) (Eq. 18).

respectively. Fig. 5 shows how symmetrization-based methods degrade compared to ParPIC as these two factors increase. Notably, symmetrization-based methods exhibit a crescent-shaped performance degradation as $\rho$ and $m_1$ increase, while ParPIC remains stable across all settings.

***Impact of number of clusters in DiSBM-CP.*** Fig. 5c shows the clustering performance of different methods when varying the number of clusters in a core-periphery structure. The following DiSBM model is considered:

$$k \text{ clusters}, m = [500, 500, ..., 500] \text{ vertices per cluster, and } \mathbf{Q} = \begin{bmatrix} 0.05 & 0.4 & \cdots & 0.4 \\ 0.02 & 0.05 & \cdots & 0.02 \\ \vdots & \vdots & \ddots & \vdots \\ 0.02 & 0.02 & \cdots & 0.05 \end{bmatrix}, \quad (18)$$

which is called DiSBM-CP($k$). In this model, one core cluster has high out-degrees toward all other non-core clusters, while the non-core clusters have low out-degrees toward each other. As the number of clusters increases, the performance of symmetrization-based methods degrades significantly, while ParPIC maintains high clustering accuracy. PIC shows performance degradation as well, but to a lesser extent compared to symmetrization-based methods. This loss is attributed to the fact that, in the model described, each non-core cluster has a low out-degree toward other non-core clusters. Thus, when the number of non-core clusters increases, the non-core clusters increasingly resemble each other structurally, making the rows of the random walk operator similar for vertices in different non-core clusters. While ParPIC mitigates this issue by the usage of its vertex measure, it is still affected by this when the number of clusters grows large.

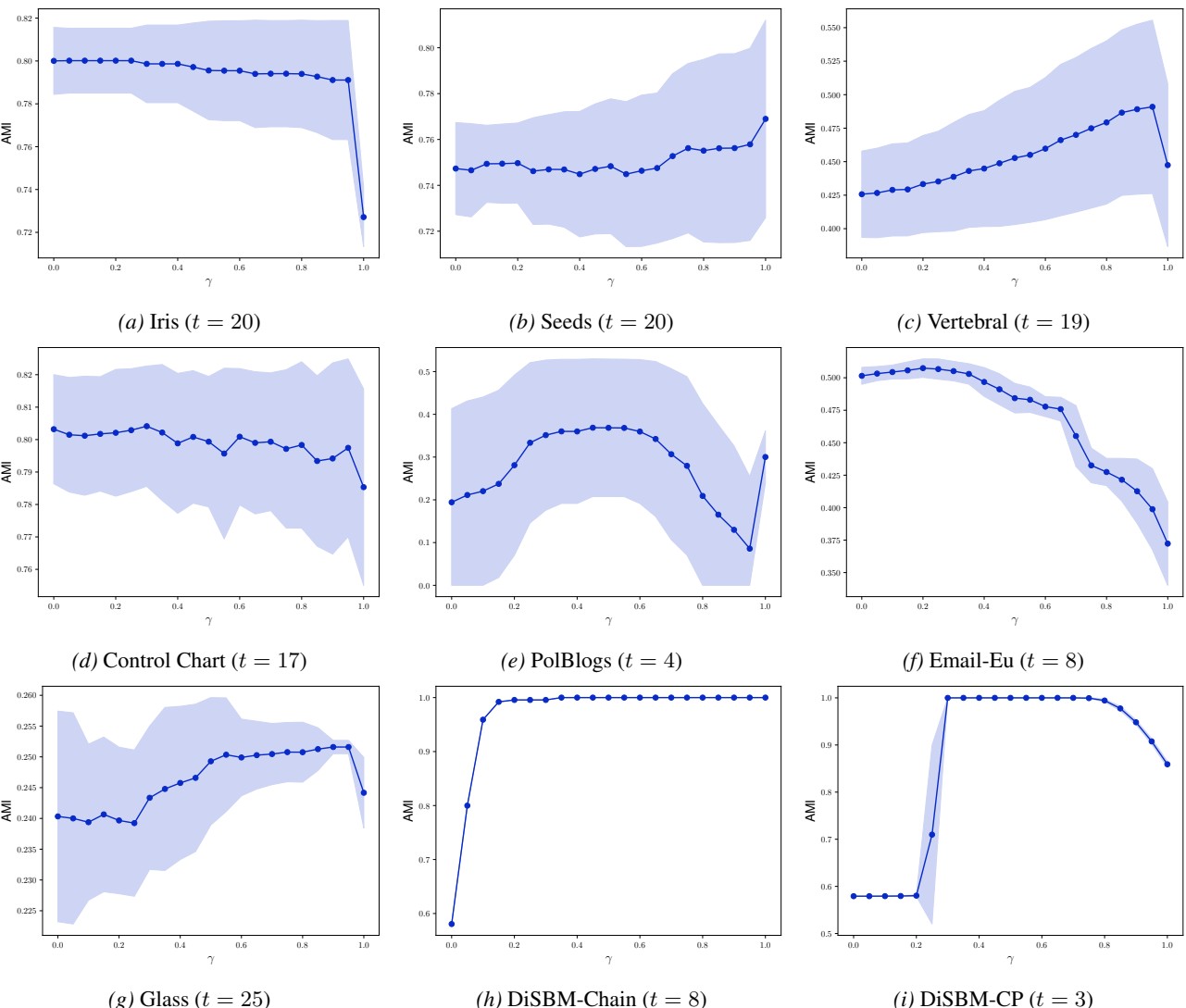

*Figure 6.* **Vertex measure parameter sensitivity.** Clustering performance (AMI) as a function of $\gamma \in [0, 1]$ across nine datasets with varying directional characteristics. The diffusion time $t$ is fixed for each dataset (values in parentheses). The performance exhibits dataset-dependent sensitivity: stable across $\gamma$ values for $K$-NN graphs (Iris, Seeds, Control Chart, Glass, Vertebral), peaked at intermediate (PolBlogs, DiSBM-CP), and improved at low $\gamma$ (Email-Eu). Default choice $\gamma = 0.5$ provides robust performance across most settings.

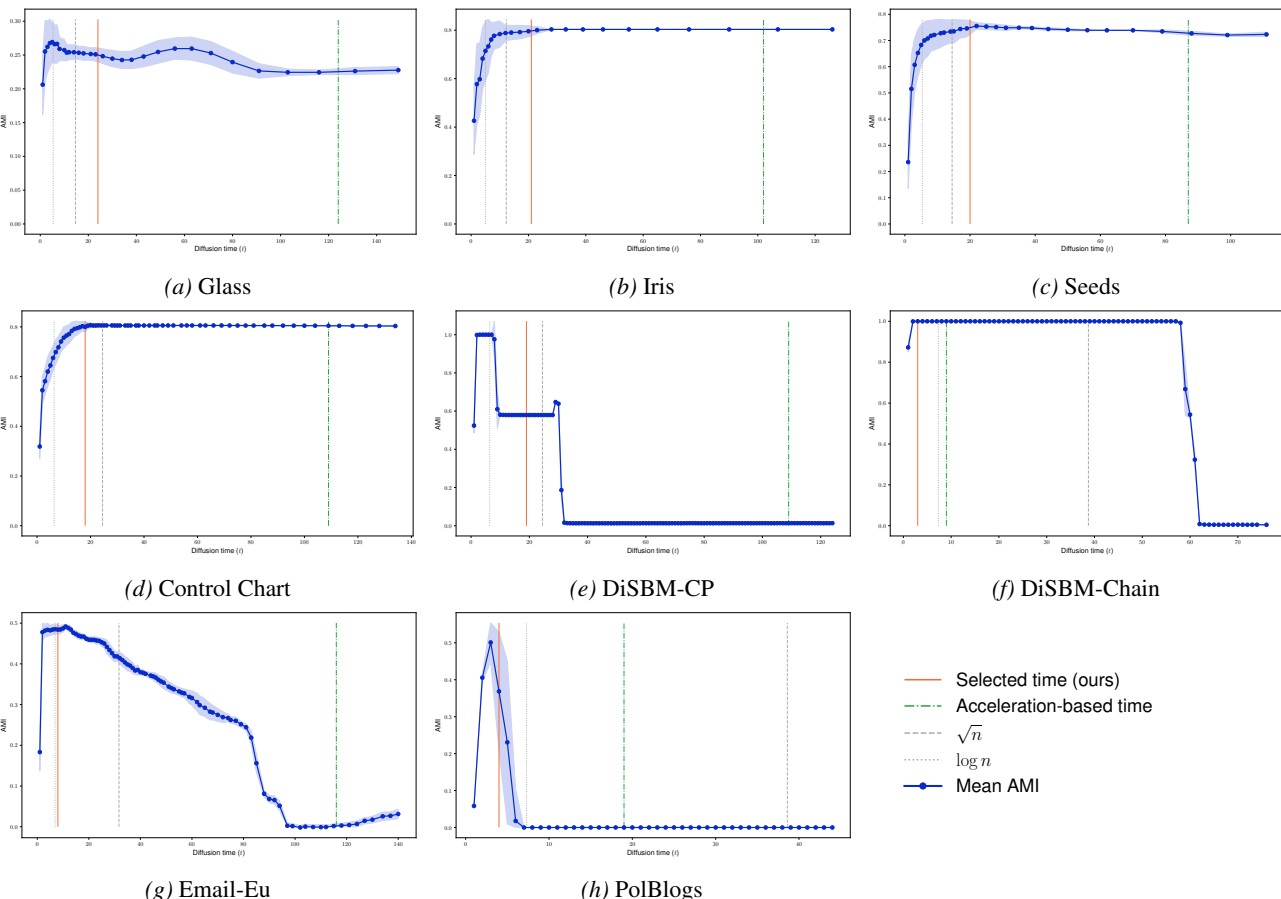

*Figure 7.* **Impact of the diffusion time** $t$ **on clustering performance and time selection methods.** Clustering performance (AMI) as a function of the diffusion time $t$ for different datasets. Different time selection strategies are indicated by vertical lines: our proposed entropy-based method (orange), an acceleration-based method (green), and two fixed-time heuristics using $\sqrt{N}$ (gray, dashed ) and $\log N$ (gray, dotted) entropy samples. The results highlight the non-monotonic relationship between diffusion time and clustering quality, with our entropy-based selection consistently achieving competitive or superior performance to other methods.

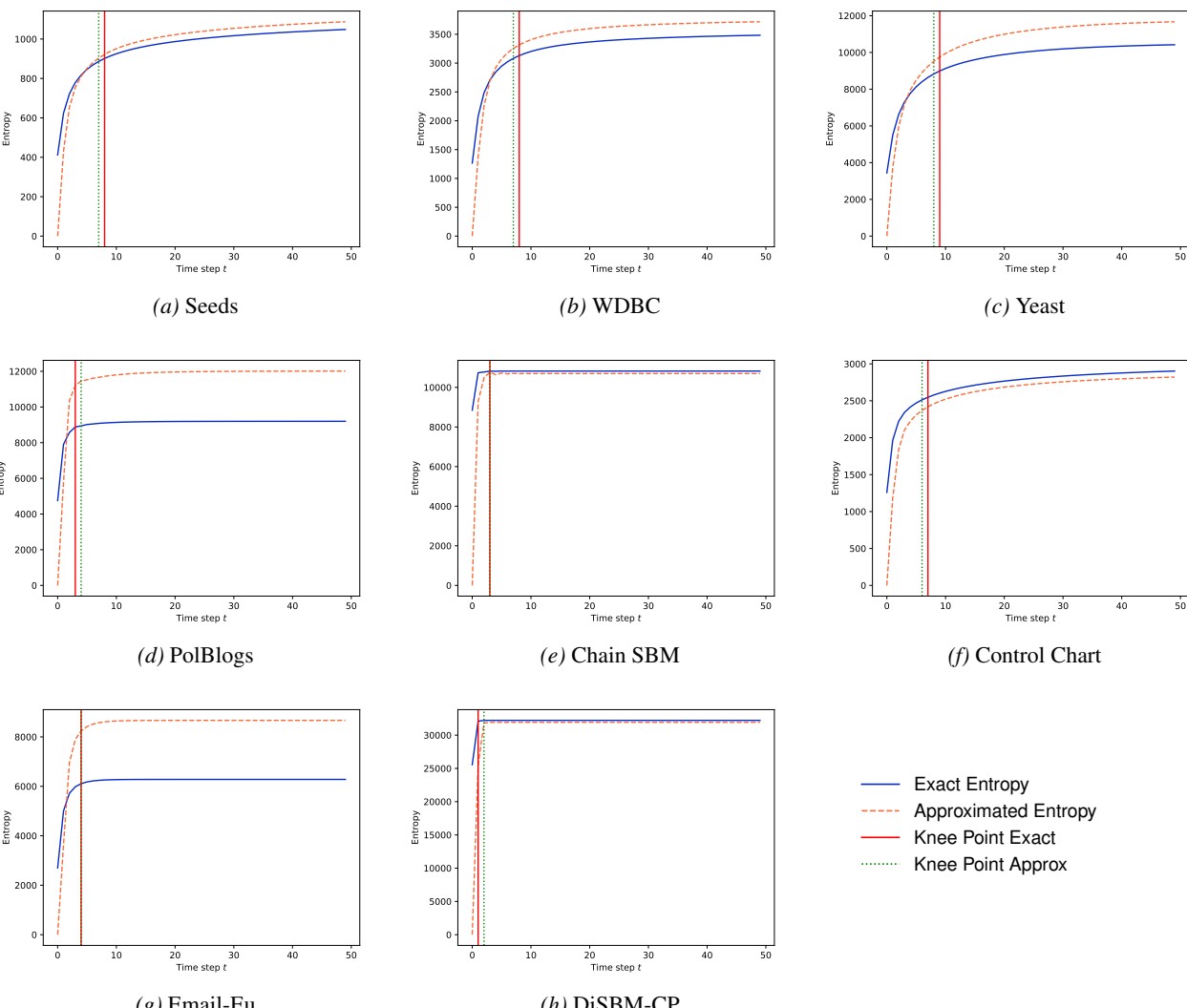

*Figure 8.* **Sampling-based entropy estimation.** Comparison of the estimated row-wise operator entropy using $\sqrt{N}$ vertex samples against the true row-wise entropy computed over the entire graph. The close alignment between the two curves demonstrates the effectiveness of the sampling approach for diffusion time selection.

# D. Sensitivity Analysis

This section presents a comprehensive sensitivity analysis examining the robustness of ParPIC to its key hyperparameters: $\gamma$, $t$, and $d$. For the vertex measure parameter $\gamma \in [0, 1]$, the balance between in-degree and out-degree and its influence on clustering performance across graphs with varying directional characteristics is assessed. For the diffusion time $t \in \mathbb{N}^*$, the proposed entropic selection criterion is validated by comparing against alternative time selection strategies (acceleration-based and graph-size-based heuristics) and examining clustering performance across a range of diffusion scales. The choice of the number of probes ($\sqrt{N}$) needed for faithful estimation of the proposed entropy functional is also validated. For the embedding dimension $d$, the trade-off between representational capacity and computational efficiency is investigated, demonstrating that moderate dimensions suffice for high-quality clustering. These analyses collectively establish the robustness of default parameter choices ($\gamma = 0.5$, entropy-selected $t$, $d = \sqrt{N}$) while providing insights into when and how parameter tuning can further improve performance.

## D.1. Vertex Measure's Parameter Sensitivity

The sensitivity of the clustering performance of ParPIC to the vertex measure parameter $\gamma$ defined in Sec. 3 is analyzed. Clustering performance is evaluated across several datasets as a function of $\gamma$, with the diffusion time $t$ fixed to a selected value for each dataset, chosen based on the elbow method described in Sec. 3.4. Fig. 6 presents the results, showing that the clustering performance of ParPIC varies across datasets. In some cases, performance is relatively insensitive to $\gamma$; this can be observed on the $K$-NN-based datasets Iris, Seeds, Vertebral, Control Chart, and Glass (Figs. 6a to 6d and 6g), though in those datasets, a value of $\gamma$ close to 1 leads to suboptimal performance. In those cases, since the graphs are built using $K$-NN, the out-degree of all nodes is the same, which impacts the proposed vertex measure. In other cases, such as PolBlogs or DiSBM-CP, performance peaks at intermediate $\gamma$ values (Figs. 6e and 6i). In Email-Eu, performance improves as $\gamma$ decreases (Fig. 6f). In DiSBM-Chain, $\gamma \geq 0.25$ gives a perfect clustering, while $\gamma = 0$ outputs a clustering that achieves 0 AMI. Overall, it is observed that choosing an intermediate value of $\gamma$ (e.g. $\gamma = 0.5$) often yields good clustering performance across various datasets, highlighting the effectiveness of balancing in-degree and out-degree information in the vertex measure, while extreme values of $\gamma$ lead to suboptimal performance. Thus, when no prior knowledge is available about the graph structure, setting $\gamma$ to an intermediate value is a reasonable default choice.

## D.2. Analysis of the Diffusion Time Parameter

The role of the diffusion time parameter $t$ in determining clustering quality is examined and the entropy-based time selection mechanism is validated. Throughout this section, the vertex measure parameter is fixed to $\gamma = 0.5$ to isolate the effect of diffusion time. Elbows in the row-wise operator entropy are identified using the KNEEDLE algorithm (Satopaa et al., 2011).

***Sensitivity to diffusion time and comparison of time selection strategies.*** Fig. 7 presents clustering performance (AMI) as a function of $t$ across diverse datasets. Four time selection strategies are compared via vertical lines: our proposed entropy-based method (orange), an acceleration-based method (Lin & Cohen, 2010) (green), and two fixed-time heuristics using $\sqrt{N}$ (dashed gray) and $\log N$ (dotted gray) entropy samples. The acceleration-based method selects $t$ when the power iteration convergence acceleration drops below a threshold ($10^{-4}$), aiming to identify when the random walk has sufficiently mixed without over-smoothing. The curves reveal a non-monotonic relationship between $t$ and clustering quality, with performance typically peaking at intermediate diffusion scales before degrading at longer times, reflecting the trade-off between under-diffusion (small $t$) and over-smoothing (large $t$). Our entropy-based selection consistently achieves competitive or superior performance across all datasets, validating the effectiveness of the operator entropy criterion in automatically identifying informative diffusion scales while selecting computationally reasonable time parameters. The acceleration-based method tends to overshoot optimal times, while the fixed-time heuristics ($\log N$, $\sqrt{N}$), though computationally inexpensive, do not adapt to graph structure and often yield suboptimal clustering. These results underscore the importance of principled, structure-aware time selection in diffusion-based clustering.

***Quality and stability of sampling-based entropy estimation.*** Fig. 8 compares the estimated row-wise operator entropy (using $\sqrt{N}$ vertex samples) against the true entropy computed over all vertices. Across datasets satisfying the theoretical assumptions (Seeds, WDBC, Yeast, Control Chart), the estimated and true entropy curves align closely, demonstrating the accuracy of the sampling approach. The entropy curves exhibit the expected monotonic increase and clear elbow patterns that enable reliable time selection.

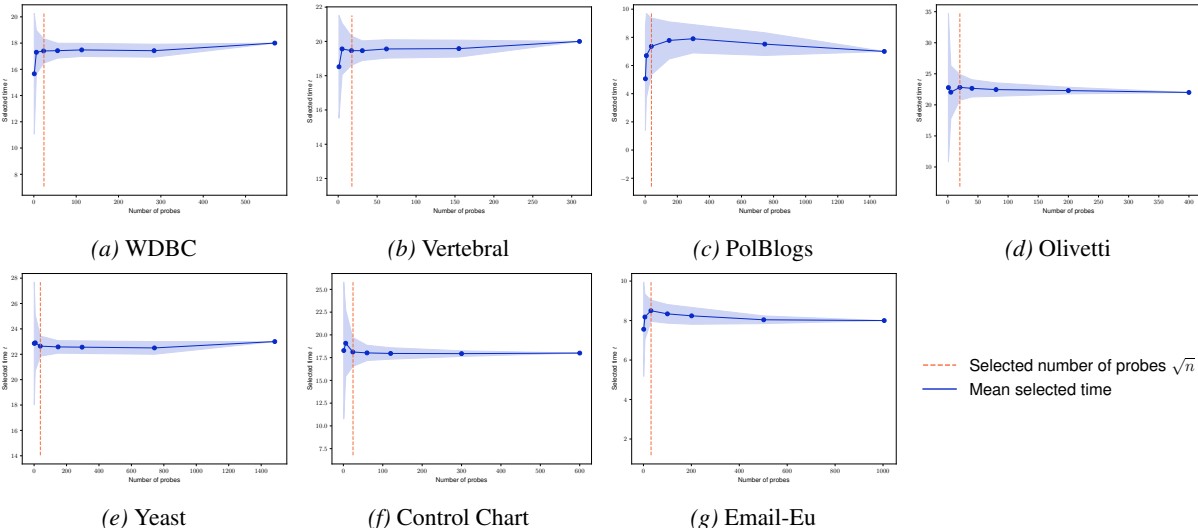

*Figure 9.* **Impact of the number of probes on the selected diffusion time.** Selected diffusion time $t$ as a function of the number of sampled vertices (probes) used in the sampling-based entropy estimation. The results demonstrate rapid convergence of the selected time as the number of probes increases, indicating that a moderate number of probes (typically $\sqrt{N}$) suffices for reliable diffusion time selection while maintaining computational efficiency.

*Convergence of time selection with respect to number of samples.* The stability of the selected diffusion time as a function of the number of vertex samples (probes) used in entropy estimation is investigated. Fig. 9 shows the selected time $t$ versus the number of probes across four datasets with varying structural properties. As the number of probes increases, the selected time rapidly stabilizes and converges to a consistent value, typically achieving near-convergence before reaching $\sqrt{N}$ samples. This convergence behavior demonstrates that a moderate number of probes suffices to capture the essential entropy dynamics for reliable time selection. The convergence is particularly rapid for well-structured datasets (WDBC, Vertebral) and requires slightly more samples for graphs with complex connectivity patterns (PolBlogs, Email-Eu). In practice, using $\sqrt{N}$ probes provides an excellent trade-off between computational efficiency and estimation accuracy, requiring fewer random walk iterations for time selection compared to exact computation.

### D.3. Embedding Dimension Sensitivity

The impact of the embedding dimension $d$ on clustering performance is evaluated to determine the trade-off between representational capacity and computational efficiency. Throughout these experiments, the vertex measure parameter is fixed to $\gamma = 0.5$ and the diffusion time $t$ is selected according to the entropic criterion defined in Sec. 3.5. Results are averaged over 50 runs with a fixed time parameter to isolate the effect of the embedding dimension.

Fig. 10 shows clustering performance (AMI) as a function of $d$ for three representative datasets. The curves exhibit a characteristic pattern: performance increases rapidly with $d$ for small dimensions, plateaus once sufficient representational capacity is achieved, and remains stable for larger values. This plateau behavior is observed across all datasets, typically occurring around $d \approx \sqrt{N}$ (indicated by orange dashed lines). Beyond this threshold, increasing $d$ provides negligible performance gains while increasing computational cost. For very small dimensions ($d < 10$), performance degrades substantially as the embedding lacks sufficient capacity to capture the diffusion structure. These results confirm that moderate embedding dimensions, specifically $d = \sqrt{N}$, provide an effective balance between clustering quality and computational efficiency. The Johnson–Lindenstrauss lemma states that for any set of $N$ points in $\mathbb{R}^D$ and any $\varepsilon \in (0, 1)$, a random projection into $\mathbb{R}^d$ with $d = \mathcal{O}(\varepsilon^{-2} \log(N))$ preserves all pairwise distances up to a multiplicative factor $(1 \pm \varepsilon)$ with high probability (Freksen, 2021).

While this suggests using an embedding dimension of the order $d = C \log N$ for some constant $C$, finding the correct constant in practice is difficult. For the moderate size of the datasets $d = \sqrt{N}$ provides a robust default choice across various settings. When dealing with substantially larger datasets, one can consider using $d = C \log N$ with a moderate constant ($C \approx 10$) to further reduce computational cost while maintaining good performance.

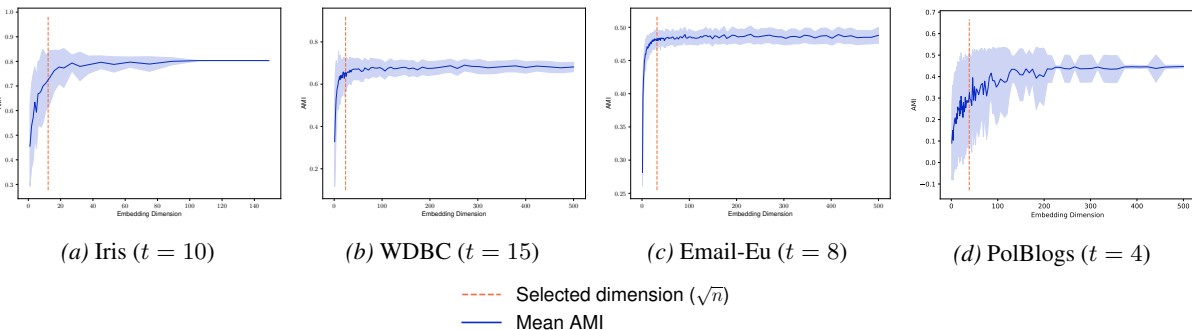

*(a)* Iris ($t = 10$)  *(b)* WDBC ($t = 15$)  *(c)* Email-Eu ($t = 8$)  *(d)* PolBlogs ($t = 4$)

- - - - Selected dimension ($\sqrt{n}$)
——— Mean AMI

*Figure 10.* **Embedding dimension sensitivity.** Clustering performance (AMI) as a function of the embedding dimension $d$. Results are computed with vertex measure parameter $\gamma = 0.5$ and diffusion time selected via the entropic criterion (Sec. 3.5). Performance is averaged over 50 runs with fixed time parameter. The vertical orange dashed line indicates the chosen dimension $d = \sqrt{N}$. Performance plateaus beyond this threshold, demonstrating that moderate embedding dimensions suffice for high-quality clustering while maintaining computational efficiency.

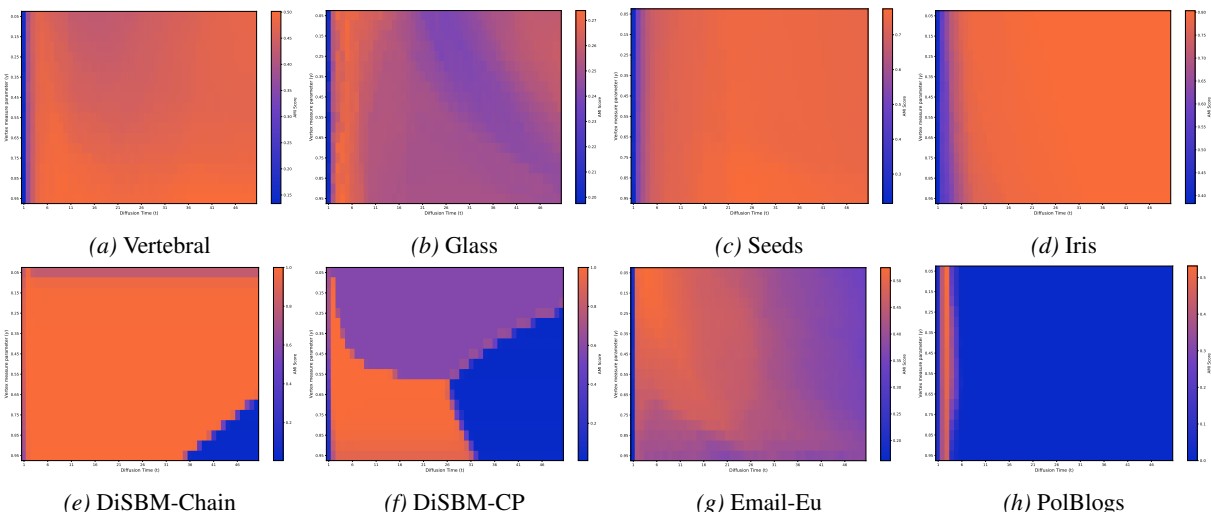

*(a)* Vertebral  *(b)* Glass  *(c)* Seeds  *(d)* Iris

*(e)* DiSBM-Chain  *(f)* DiSBM-CP  *(g)* Email-Eu  *(h)* PolBlogs

*Figure 11.* **Joint vertex measure parameter and diffusion time sensitivity analysis.** Clustering performance (AMI) as a function of the vertex measure parameter $\gamma$ and diffusion time $t$ for additional datasets. The results indicate that the choice of $\gamma$ and $t$ can significantly influence clustering performance, with different datasets exhibiting varying sensitivities.

## D.4. Joint Analysis and Conclusions on Parameter Sensitivity

***Joint sensitivity of $\gamma$ and $t$.*** The joint sensitivity of clustering performance to both the vertex measure parameter $\gamma$ and diffusion time $t$ is analyzed. Fig. 11 shows clustering performance (AMI) as a function of $(\gamma, t)$ for some benchmark datasets, with $\gamma \in \{0.05, 0.1, ..., 0.95\}$ and $t \in \{1, 2, ..., 50\}$. The different datasets exhibit varying sensitivities to these parameters. In some cases, such as Seeds and Iris (Figs. 11c and 11d), performance is relatively stable across a range of $\gamma$ and $t$ values, indicating robustness to parameter choices. In other cases, performance peaks sharply at specific $(\gamma, t)$ combinations, highlighting the importance of careful tuning to capture the underlying community structure effectively. Overall, this joint analysis underscores the interplay between vertex measure design and diffusion dynamics in shaping clustering outcomes on directed graphs.

***Conclusions on sensitivity analyses.*** From the sensitivity analyses conducted on the vertex measure parameter $\gamma$, diffusion time $t$, and embedding dimension $d$, several conclusions are drawn regarding the robustness and adaptability of the proposed method, ParPIC. First, the vertex measure parameter $\gamma$ plays a crucial role in balancing the influence of in-degree and out-degree in the clustering process. While $\gamma = 0.5$ serves as a safe default choice, allowing for equal weighting, the sensitivity analysis reveals that certain datasets benefit from specific $\gamma$ values, particularly in scenarios where directionality is a key factor in community structure. Second, the diffusion time $t$ can impact clustering performance, as well as computational

efficiency. The proposed entropy-based time selection effectively estimates diffusion times yielding competitive clustering results, at a reasonable computational cost. Finally, the embedding dimension $d$ shows a degree of robustness, with the method maintaining strong performance across a range of dimensions, especially when $d \geq \sqrt{N}$, where a plateau of values in the AMI scores is often observed. Overall, these experiments highlight the method's flexibility and effectiveness across diverse directed graph structures, while also providing practical guidelines for parameter selection to optimize clustering performance.

# E. Proofs

***Proof of Proposition 3.2.*** For completeness, the proof begins by showing that the matrix $\mathbf{P}_{(\nu)}$ is stochastic.

$$\sum_{j=1}^{N} \mathbf{P}_{(\nu)}(i,j) = \sum_{j=1}^{N} \frac{\nu(i)\mathbf{P}_{ij} + \nu(j)\mathbf{P}_{ji}}{\nu(i) + \xi(i)}$$

$$= \frac{\nu(i)\sum_{j=1}^{N}\mathbf{P}_{ij} + \sum_{j=1}^{N}\nu(j)\mathbf{P}_{ji}}{\nu(i) + \xi(i)}$$

$$= \frac{\nu(i) + \xi(i)}{\nu(i) + \xi(i)} = 1.$$

The last line is due to the fact that $\xi(i) = \sum_{j=1}^{N} \nu(j)\mathbf{P}_{ji}$, thus, $\mathbf{P}_{(\nu)}$ is a stochastic matrix.

Next, we prove that for any positive vertex measure $\nu$, $\mathbf{P}_{(\nu)}$ is reversible. Using the definition of $\mathbf{P}_{(\nu)}$, it holds that:

$$(\mathbf{D}_\nu + \mathbf{D}_\xi)\mathbf{P}_{(\nu)} = \mathbf{D}_\nu \mathbf{P} + \mathbf{P}^\mathsf{T}\mathbf{D}_\nu = (\mathbf{D}_\nu + \mathbf{D}_\xi)\mathbf{P}_{(\nu)}^\mathsf{T},$$

which is symmetric since $\mathbf{D}_\nu \mathbf{P} + \mathbf{P}^\mathsf{T}\mathbf{D}_\nu$ is symmetric. Thus, $\mathbf{P}_{(\nu)}$ is reversible with respect to the measure $\nu + \xi$, and if it exists, the stationary distribution $\pi_{(\nu)}$ satisfies $\pi_{(\nu)} \propto \nu + \xi$, where $\pi_{(\nu)}$ is the ergodic distribution associated to $\mathbf{P}_{(\nu)}$. Next, we prove the irreducibility and aperiodicity properties. Notice that for any $i, j \in \{1, ..., N\}$:

$$\mathbf{P}_{(\nu)}(i,j) = \frac{\nu(i)\mathbf{P}_{ij} + \nu(j)\mathbf{P}_{ji}}{\nu(i) + \xi(i)} > 0 \quad \Longleftarrow \quad \mathbf{P}_{ij} > 0 \text{ or } \mathbf{P}_{ji} > 0.$$

In particular, if there exists an undirected path between any two vertices in the digraphs, then there exists a directed path between any two vertices in the graph associated to $\mathbf{P}_{(\nu)}$, thus $\mathbf{P}_{(\nu)}$ is irreducible, when the original digraph is weakly connected. Moreover, if $\mathbf{P}_{ii} > 0$, then $\mathbf{P}_{(\nu)}(i,i) > 0$, thus $\mathbf{P}_{(\nu)}$ is aperiodic, when the original digraph has at least one self-loop.

For the continuity property, let $\nu, \mu \in \mathbb{R}^N$ be two vertex measures, and denote $\xi = \mathbf{P}^\mathsf{T}\nu$ and $\xi' = \mathbf{P}^\mathsf{T}\mu$. We have that:

$$\left\|\mathbf{P}_{(\nu)} - \mathbf{P}_{(\mu)}\right\| = \left\|(\mathbf{D}_\nu + \mathbf{D}_\xi)^{-1}(\mathbf{D}_\nu\mathbf{P} + \mathbf{P}^\mathsf{T}\mathbf{D}_\nu) - (\mathbf{D}_\mu + \mathbf{D}_{\xi'})^{-1}(\mathbf{D}_\mu\mathbf{P} + \mathbf{P}^\mathsf{T}\mathbf{D}_\mu)\right\|$$

$$\leq \left\|(\mathbf{D}_\nu + \mathbf{D}_\xi)^{-1} - (\mathbf{D}_\mu + \mathbf{D}_{\xi'})^{-1}\right\| \cdot \left\|\mathbf{D}_\nu\mathbf{P} + \mathbf{P}^\mathsf{T}\mathbf{D}_\nu\right\|$$

$$+ \left\|(\mathbf{D}_\mu + \mathbf{D}_{\xi'})^{-1}\right\| \cdot \left\|(\mathbf{D}_\nu - \mathbf{D}_\mu)\mathbf{P} + \mathbf{P}^\mathsf{T}(\mathbf{D}_\nu - \mathbf{D}_\mu)\right\|.$$

Using the resolvent identity[1] to $\left\|(\mathbf{D}_\nu + \mathbf{D}_\xi)^{-1} - (\mathbf{D}_\mu + \mathbf{D}_{\xi'})^{-1}\right\|$, we obtain:

$$\left\|(\mathbf{D}_\nu + \mathbf{D}_\xi)^{-1} - (\mathbf{D}_\mu + \mathbf{D}_{\xi'})^{-1}\right\| \leq \left\|(\mathbf{D}_\nu + \mathbf{D}_\xi)^{-1}(\mathbf{D}_\mu + \mathbf{D}_{\xi'})^{-1}\right\|\left\|(\mathbf{D}_\nu - \mathbf{D}_\mu) + (\mathbf{D}_\xi - \mathbf{D}_{\xi'})\right\|$$

$$\leq \left\|(\mathbf{D}_\nu + \mathbf{D}_\xi)^{-1}(\mathbf{D}_\mu + \mathbf{D}_{\xi'})^{-1}\right\|\left(1 + \|\mathbf{P}\|\right)\|\nu - \mu\|.$$

Where the last inequality comes from the fact that $\xi - \xi' = \mathbf{P}^\mathsf{T}(\nu - \mu)$, and thus $\left\|\xi - \xi'\right\| \leq \|\mathbf{P}\|\|\nu - \mu\|$. For the other term, we can bound using the triangle inequality and the fact that $\mathbf{D}_\nu$ and $\mathbf{D}_\mu$ are diagonal matrices:

$$\left\|\mathbf{D}_\nu\mathbf{P} + \mathbf{P}^\mathsf{T}\mathbf{D}_\nu\right\| \leq 2\|\mathbf{P}\|\|\nu\|, \quad \text{and} \quad \left\|(\mathbf{D}_\nu - \mathbf{D}_\mu)\mathbf{P} + \mathbf{P}^\mathsf{T}(\mathbf{D}_\nu - \mathbf{D}_\mu)\right\| \leq 2\|\mathbf{P}\|\|\nu - \mu\|.$$

---

[1] for $\mathbf{A}, \mathbf{B}$ invertible matrices, $\mathbf{A}^{-1} - \mathbf{B}^{-1} = \mathbf{A}^{-1}(\mathbf{B} - \mathbf{A})\mathbf{B}^{-1}$, this can be seen by developing the right-hand side equality.

Combining those inequalities, we obtain that the difference $\left\| \mathbf{P}_{(\nu)} - \mathbf{P}_{(\mu)} \right\|$ vanishes as $\|\nu - \mu\| \to 0$, and thus that $\mathbf{P}_{(\nu)}$ is continuous with respect to $\nu$.

For the last point, assume that the underlying graph is undirected and ergodic. We show that $\mathbf{P}_{(\pi)} = \mathbf{P}$. Let $\nu = \pi$, where $\pi$ is the ergodic distribution of the natural random walk. Then $\xi = \mathbf{P}^{\mathsf{T}} \nu = \pi$, so $\nu + \xi = 2\pi$ and therefore $\mathbf{D}_\nu + \mathbf{D}_\xi = 2\mathbf{D}_\pi$. Moreover, in the undirected setting, detailed balance holds:

$$\mathbf{D}_\pi \mathbf{P} = \mathbf{P}^{\mathsf{T}} \mathbf{D}_\pi,$$

since $\pi(i) \propto d(i)$ and $\mathbf{P}(i,j) = \mathbf{W}(i,j)/d(i)$ with $\mathbf{W}$ symmetric. Hence,

$$\mathbf{D}_\pi \mathbf{P} + \mathbf{P}^{\mathsf{T}} \mathbf{D}_\pi = 2\mathbf{D}_\pi \mathbf{P}.$$

Substituting into the definition of $\mathbf{P}_{(\nu)}$ gives

$$\mathbf{P}_{(\pi)} = (\mathbf{D}_\nu + \mathbf{D}_\xi)^{-1}(\mathbf{D}_\nu \mathbf{P} + \mathbf{P}^{\mathsf{T}} \mathbf{D}_\nu) = (2\mathbf{D}_\pi)^{-1}(2\mathbf{D}_\pi \mathbf{P}) = \mathbf{P}.$$

Therefore, the P-RW operator coincides with the natural random walk operator in this case. ∎

For completeness, we also provide the proof of the following property, linking the parametrized diffusion distance to the diffusion embedding space.

***Property.*** The parametrized diffusion distance associated to the parametrized random walk can be expressed as the Euclidean distance in the diffusion embedding space. If $\mathbf{P}_{(\nu)} = \mathbf{\Phi}_{(\nu)} \mathbf{\Lambda}_{(\nu)} \mathbf{\Phi}_{(\nu)}^{-1}$ is the eigen-decomposition of $\mathbf{P}_{(\nu)}$, and $\Psi_{t,(\nu)} = \mathbf{\Phi}_{(\nu)} \mathbf{\Lambda}_{(\nu)}^t$, then:

$$\mathcal{D}_{t,(\nu)}^2(i,j) = \left\| \Psi_{t,(\nu)}^{\mathsf{T}} (\delta_i - \delta_j) \right\|_2^2. \tag{19}$$

*Proof.* Using the definition of the parametrized random walk operator $\mathbf{P}_{(\nu)}$, we have that if $\nu > 0$ component-wise, $\mathbf{P}_{(\nu)}$ is reversible with respect to $\pi_{(\nu)} \propto \nu + \xi$ and thus diagonalizable with real eigenvalues. It admits the spectral decomposition $\mathbf{P}_{(\nu)} = \mathbf{\Phi}_{(\nu)} \mathbf{\Lambda}_{(\nu)} \mathbf{\Phi}_{(\nu)}^{-1}$, where $\mathbf{\Phi}_{(\nu)}$ is the matrix of eigenvectors and $\mathbf{\Lambda}_{(\nu)} = \operatorname{diag}(\lambda_1, ..., \lambda_N)$ is the diagonal matrix of eigenvalues. The eigenvectors are orthonormal in $\ell_2(\pi_{(\nu)})$, i.e. $\mathbf{\Phi}_{(\nu)}^{\mathsf{T}} \mathbf{D}_{\pi_{(\nu)}} \mathbf{\Phi}_{(\nu)} = I$ so that $\mathbf{\Phi}_{(\nu)}^{-1} = \mathbf{\Phi}_{(\nu)}^{\mathsf{T}} \mathbf{D}_{\pi_{(\nu)}}$. The $t$-th power of $\mathbf{P}_{(\nu)}$ can be expressed as $\mathbf{P}_{(\nu)}^t = \mathbf{\Phi}_{(\nu)} \mathbf{\Lambda}_{(\nu)}^t \mathbf{\Phi}_{(\nu)}^{-1}$. Starting from the definition of the parametrized diffusion distance (Eq. 4):

$$\mathcal{D}_{t,(\nu)}^2(i,j) = \sum_{l=1}^N \frac{1}{\pi_{(\nu)}(l)} \left( \mathbf{P}_{(\nu)}^t(i,l) - \mathbf{P}_{(\nu)}^t(j,l) \right)^2.$$

By the spectral decomposition, we have component-wise $\mathbf{P}_{(\nu)}^t(i,l) = \sum_{m=1}^N \lambda_m^t \, \mathbf{\Phi}_{(\nu)}(i,m) \, \mathbf{\Phi}_{(\nu)}(l,m) \pi_{(\nu)}(l)$, therefore:

$$\mathcal{D}_{t,(\nu)}^2(i,j) = \sum_{l=1}^N \frac{1}{\pi_{(\nu)}(l)} \left( \sum_{m=1}^N \lambda_m^t \, \pi_{(\nu)}(l) \mathbf{\Phi}_{(\nu)}(l,m) \left( \mathbf{\Phi}_{(\nu)}(i,m) - \mathbf{\Phi}_{(\nu)}(j,m) \right) \right)^2.$$

$$= \sum_{m=1}^N \sum_{m'=1}^N \lambda_m^t \lambda_{m'}^t \left( \mathbf{\Phi}_{(\nu)}(i,m) - \mathbf{\Phi}_{(\nu)}(j,m) \right) \left( \mathbf{\Phi}_{(\nu)}(i,m') - \mathbf{\Phi}_{(\nu)}(j,m') \right)$$

$$\times \sum_{l=1}^N \mathbf{\Phi}_{(\nu)}(l,m) \, \mathbf{\Phi}_{(\nu)}(l,m') \pi_{(\nu)}(l).$$

Due to the orthonormality of the eigenvectors in $\ell_2(\pi_{(\nu)})$: $(\mathbf{\Phi}_{(\nu)}(l,m) \, \mathbf{\Phi}_{(\nu)}(l,m')) \pi_{(\nu)}(l) = 0$ if $m \neq m'$ and $1$ if $m = m'$.

The last sum simplifies:

$$\mathcal{D}_{t,(\nu)}^2(i,j) = \sum_{m=1}^{N} \lambda_m^{2t} \Big( \mathbf{\Phi}_{(\nu)}(i,m) - \mathbf{\Phi}_{(\nu)}(j,m) \Big)^2$$

$$= \sum_{m=1}^{N} \Big( \lambda_m^t \mathbf{\Phi}_{(\nu)}(i,m) - \lambda_m^t \mathbf{\Phi}_{(\nu)}(j,m) \Big)^2$$

$$= \left\| \Psi_{t,(\nu)}^{\mathsf{T}} \delta_i - \Psi_{t,(\nu)}^{\mathsf{T}} \delta_j \right\|_2^2 = \left\| \Psi_{t,(\nu)}^{\mathsf{T}} (\delta_i - \delta_j) \right\|_2^2,$$

where $\Psi_{t,(\nu)}(i,m) = \lambda_m^t \mathbf{\Phi}_{(\nu)}(i,m)$ defines the diffusion map embedding. This concludes the proof. ∎

***Proof of Proposition* 3.5.** First, we can start by noting that $\mathcal{H}(t)$ is indeed positive, $\mathbf{P}_{(\nu)}(i,j) \in [0,1]$ so $\log(\mathbf{P}_{(\nu)}(i,j)) \le 0$, $\mathcal{H}_i(t) \ge 0$ and $\mathcal{H}(t) = \frac{1}{N} \sum_i \mathcal{H}_i(t) \ge 0$. Then, we continue by showing that the entropy $\mathcal{H}(t)$ is non-decreasing with respect to $t$. Using the definition, we have that $\mathbf{P}_{(\nu)}^{t+1}(i,j) = \sum_{k=1}^{N} \mathbf{P}_{(\nu)}^t(i,k) \times \mathbf{P}_{(\nu)}(k,j)$. Note that the function $f(x) = -x \log x$ for $x \in [0,1]$ is concave (since $f''(x) = -1/x < 0$). By Jensen's inequality applied to the concave function $f$:

$$\mathcal{H}_i(t+1) = -\sum_j \mathbf{P}_{(\nu)}^{t+1}(i,j) \log \mathbf{P}_{(\nu)}^{t+1}(i,j)$$

$$= \sum_j f\left( \sum_k \mathbf{P}_{(\nu)}^t(i,k) \mathbf{P}_{(\nu)}(k,j) \right)$$

$$\ge \sum_j \sum_k \mathbf{P}_{(\nu)}(k,j) \cdot f\left( \mathbf{P}_{(\nu)}^t(i,k) \right)$$

$$= \sum_k f\left( \mathbf{P}_{(\nu)}^t(i,k) \right) \sum_j \mathbf{P}_{(\nu)}(k,j)$$

$$= -\sum_k \mathbf{P}_{(\nu)}^t(i,k) \log \left( \mathbf{P}_{(\nu)}^t(i,k) \right) = \mathcal{H}_i(t),$$

where we used the fact that $\sum_j \mathbf{P}_{(\nu)}(k,j) = 1$ since $\mathbf{P}_{(\nu)}$ is stochastic. Moreover, as $t \to \infty$, each row of $\mathbf{P}_{(\nu)}^t$ converges to the stationary distribution $\pi_{(\nu)}$, thus:

$$\lim_{t \to \infty} \mathcal{H}_i(t) = -\sum_j \pi_{(\nu)}(j) \log \pi_{(\nu)}(j) \quad \text{and} \quad \lim_{t \to \infty} \mathcal{H}(t) = -\sum_j \pi_{(\nu)}(j) \log \pi_{(\nu)}(j) = C(\pi_{(\nu)}).$$

This concludes the proof. ∎

***Proof of Proposition* 3.6.** Recall that $\hat{\mathcal{H}}(t) = \frac{N}{n} \sum_{r=1}^{n} \mathcal{H}_{i_r}(t)$, where $i_1, ..., i_n$ are sampled uniformly at random without

replacement from $\{1, ..., N\}$. It holds that:

$$\mathbb{E}\left[\hat{\mathcal{H}}(t)\right] = \frac{N}{n}\mathbb{E}\left[\sum_{r=1}^{n}\mathcal{H}_{i_r}(t)\right]$$

$$= \frac{N}{n}\sum_{i=1}^{N}\mathbb{E}[\mathcal{H}_i(t)\mathbb{1}\{i \in S\}]$$

$$= \frac{N}{n}\sum_{i=1}^{N}\mathcal{H}_i(t)\mathbb{E}[\mathbb{1}\{i \in S\}]$$

$$= \frac{N}{n}\sum_{i=1}^{N}\mathcal{H}_i(t)\mathbb{P}(i \in S)$$

$$= \frac{N}{n}\sum_{i=1}^{N}\mathcal{H}_i(t)\frac{n}{N}$$

$$= \sum_{i=1}^{N}\mathcal{H}_i(t) = \mathcal{H}(t).$$

This means that $\hat{\mathcal{H}}(t)$ is an unbiased estimator of $\mathcal{H}(t)$. For the variance, we have the following:

$$\mathrm{Var}(\hat{\mathcal{H}}(t)) = \mathrm{Var}\left(\frac{N}{n}\sum_{r=1}^{n}\mathcal{H}_{i_r}(t)\right)$$

$$= \frac{N^2}{n^2}\mathrm{Var}\left(\sum_{r=1}^{n}\mathcal{H}_{i_r}(t)\right)$$

$$= \frac{N^2}{n^2}\left[n\mathrm{Var}(\mathcal{H}_{i_1}(t)) + n(n-1)\mathrm{Cov}(\mathcal{H}_{i_1}(t), \mathcal{H}_{i_2}(t))\right]$$

$$= \frac{N^2}{n}\left[\mathrm{Var}(\mathcal{H}_{i_1}(t)) + (n-1)\mathrm{Cov}(\mathcal{H}_{i_1}(t), \mathcal{H}_{i_2}(t))\right]$$

$$= \frac{N^2}{n}\frac{N-n}{N-1}\mathrm{Var}(\mathcal{H}_{i_1}(t)). \tag{20}$$

Here, we used the fact that under a simple random sampling without replacement,

$$\mathrm{Cov}(\mathcal{H}_{i_1}(t), \mathcal{H}_{i_2}(t)) = -\frac{\mathrm{Var}(\mathcal{H}_{i_1}(t))}{N-1},$$

which follows from the exchangeability of the sampled variables together with the identity $\mathrm{Var}\left(\sum_{r=1}^{N}\mathcal{H}_{i_r}(t)\right) = 0$. Using Proposition 3.5, we have that $\mathcal{H}_{i_r}(t) \in [0, C(\pi_{(\nu)})]$, thus, by Popoviciu's inequality on the variance, we have that $\mathrm{Var}(\mathcal{H}_{i_r}(t)) \leq \frac{C(\pi_{(\nu)})^2}{4}$. Eq. 20 gives:

$$\mathrm{Var}(\hat{\mathcal{H}}(t)) \leq \frac{N^2}{n}\frac{N-n}{(N-1)}\frac{C(\pi_{(\nu)})^2}{4}.$$

This allows to use a Chebyshev inequality so that:

$$\mathbb{P}\left(|\hat{\mathcal{H}}(t) - \mathcal{H}(t)| \geq \epsilon\right) \leq \frac{N^2}{n}\frac{N-n}{(N-1)}\frac{C(\pi_{(\nu)})^2}{4\epsilon^2}.$$

This implies that for any $\eta > 0$, with probability at least $1 - \eta$:

$$|\hat{\mathcal{H}}(t) - \mathcal{H}(t)| \leq \frac{N}{\sqrt{n}}\frac{C(\pi_{(\nu)})}{2}\sqrt{\frac{N-n}{(N-1)\eta}}.$$

∎

