# OpenReview forum: "Parametrized Power-Iteration Clustering for Directed Graphs"
_ICML.cc/2026/Conference — ICML 2026 regular_

### Official Review · Reviewer_zxFJ · 2026-02-25

**Soundness:** 4
**Presentation:** 3
**Significance:** 3
**Originality:** 2
**Overall Recommendation:** 4
**Confidence:** 4

**Summary:**

This paper proposes Parametrized Power-Iteration Clustering (ParPIC) for clustering weakly connected directed graphs. The method proceeds as follows. Firstly, a parametrized random walk operator $P_{(\nu)} \in \mathbb{R}^{N \times N}$ is defined, depending on the transition matrix $P$ of the original graph and an auxiliary measure $\nu$ on the vertices. Secondly, $d$ random projections of the matrix exponential $P_{(\nu)}^t$ are computed, yielding $Z_{(\nu)}^t$. Finally, the rows of $Z_{(\nu)}^t$ are clustered using the $k$-means algorithm. Some recommendations are given for choosing the diffusion time parameter $t$, the vertex measure $\nu$ and the approximation dimension $d$. Certain theoretical properties of the operator $P_{(\nu)}^t$ are established. Comprehensive numerical experiments are presented, in which the statistical performance is seen to be competitive with existing state-of-the-art approaches, while the computational burden is substantially smaller.

**Compliance With Llm Reviewing Policy:**

Affirmed.

**Final Justification:**

The authors have addressed my concerns and I am recommending 5: accept. The soundness, presentation and significance of the work are all good, and while the originality is not especially great, I do think it merits publication on account of its improvements (methodological and empirical) over existing approaches.

**Key Questions For Authors:**

See the discussion above. The main points are about improving the explanation of why the method is proposed, and for what reasons one should expect it to perform well.

**Limitations:**

Yes

**Strengths And Weaknesses:**

The paper proposes a sensible generalisation of existing approaches (see, in particular, Lin and Cohen 2020, ICML, "Power iteration clustering") for directed graph clustering. The proposed methodology retains the directed information from the original graph while fixing technical issues associated with diffusion on directed graphs; in particular, the transition matrix may not have real eigenvalues in general. In terms of soundness, the paper generally seems correct; I have checked the proofs and the empirical results are believable. The presentation is generally of a reasonably high standard; the writing is usually easy to follow. See the itemized points below for some minor issues. The work has the potential to be moderately significant. Clustering for directed graphs is a central problem in contemporary network analysis, and the improved statistical and computational performance above existing methods is valuable. The paper would much benefit from some deeper discussion on why this method is proposed and what the reasons are for its good performance. In particular, it is not clear to me why this approach is preferable to PIC, PR-PIC and S-PIC; indeed the empirical results are often very similar. Further, the explanations given in Section 3.1 do not give a full picture about what the operator $P_{(\nu)}$ is really capturing, and how it is affected by the quantities $\nu$ and $P$. Why should one expect this approach to be better than simply symmetrising the graph or applying power iterations directly to $P$? Originality is present but not especially substantial; the proposed method is an extension of existing algorithms, although it does seem to be a practical and useful one.

Other comments:

- Section 3.4: It is not clear what one is trying to optimise while choosing the diffusion time. As such, it is difficult to evaluate how sensible the proposed method is. Indeed, while the elbow heuristic probably works well in practice, could a more explicit penalised version of the entropy instead be minimised?
- Section 3.5: Is there any theory for how large $d$ must be taken to ensure that the low-dimensional approximation is sufficiently good? It seems that this convergence should be faster when the eigenvalues of $P_{(nu)}$ decay to zero quickly.

- Equation 10: It should not be hard to prove that $\hat{\mathcal{H}}(t)$ converges to $\mathcal{H}(t)$ at a rate no slower than $N/\sqrt{n}$, by Chebyshev's inequality.

Minor comments:

- Column 1, Line 70: Define $N$ before its first usage.
- Column 1, Lines 98, 100: Missing definition of $d_\mathrm{out}$ and $d_\mathrm{in}$; are these weighted vertex degrees?
- Column 1, Lines 99, 100: Too many parentheses here.
- Column 1, Equation 1: Define $\pi(k)$ before its first usage.
- Column 1, Equation 2: It might be helpful to mention here that the eigenvalues of $P$ are real for an undirected graph.
- Proposition 3.2: It is already assumed that $\nu(i) > 0$; this does not need to be a condition here.
- Column 2, Line 132: This sentence says "reversible" twice.
- Column 1, Line 191: What is $k$ here? It is not mentioned in Definition 3.4.
- Table 2: Should be $N$, not $n$ here I think (number of vertices).

---

> ### Author Rebuttal · Authors · 2026-03-30
>
> We would like to thank the reviewer for their positive and constructive feedback.  We are pleased that the paper was recognized as well-motivated and technically sound, and that the empirical evaluation, particularly the ablation study, was found convincing. We also appreciate that the recognition of the method’s value to the community and the usefulness of the proposed entropic criterion. We address the main comments below, and will make sure to include the suggested improvements in the revised version of the paper.
>
> **Why is the proposed approach preferable to other PIC variants.** While the empirical results on $K$-NN based datasets show rather similar results between the four methods, the results on the network datasets exhibit stronger performance gaps. This is particularly evident in the context of the Core-Periphery DiSBM, which has also been investigated more thoroughly in Appendix C (see Fig.5). Both DiSBM examples show that the symmetrization of the operator leads to worse clustering performance. Standard PIC fails in the presence of sinks or sources, the disruption of the random-walk dynamics potentially results in a non-unique or degenerate stationary distribution, which is particularily visible in the DiSBM Chain. PR-PIC addresses these issues, but only at the cost of densifying the operator, which limits scalability to larger graphs. S-PIC maintains sparsity, but fails to capture essential graph structure as it merely averages the natural random walk operator and its transpose. For example, in a core-periphery graph, symmetrization treats core-core and core-periphery edges identically, whereas the proposed approach distinguishes between them.
>
> ---
>
> **Diffusion time selection.** We agree that it is important to clarify the intuition behind the used elbow (or knee) heuristic. The idea is to select the diffusion time $t$ such as it is not 'too small' nor 'too big'. The knee heuristic allows balancing these two effects by looking at the curve of entropy as a function of $t$, and selecting the point where the curve starts to flatten, which corresponds to a good trade-off between noise and signal. This is a standard approach in the diffusion literature (e.g. [Debaussart-Joniec & Kalogeratos, 2025](https://arxiv.org/pdf/2512.01484); [Kuchroo et al., 2021](https://icml-compbio.github.io/2021/papers/WCBICML2021_paper_35.pdf)). In practice, we use the `Kneed` algorithm, which detects elbow by finding the point that is the closest to the diagonal, after a normalization of the curve ([Satopaa et al., 2011](https://ieeexplore.ieee.org/document/5961514)), and as such, the definition of the found elbows is 'explicit', and not hand-crafted.
>
> ---
>
> **Embedding dimension selection.** Our use of random projection is motivated by the Johnson–Lindenstrauss lemma. Since the objective is to cluster the rows of $P_{(\nu)}$, the lemma guarantees that pairwise distances between these rows are approximately preserved under a suitable random projection. In particular, for the projected matrix $Z_t$ we have:
> $$
>  \|Z_t(i, \cdot) - Z_t(j, \cdot)\| \approx \|P^t(i, \cdot) - P^t(j, \cdot)\|
> $$
> with high probability, provided the projection dimension is sufficiently large. In practice, we found that a projection dimension on the order of $\sqrt{N}$ works well. From a theoretical standpoint, however, a dimension on the order of $O(\log{N})$ (up to a multiplicative constant depending on the desired accuracy) is sufficient to ensure the JL guarantee. This allows us to efficiently embed the data in a lower-dimensional space without explicitly computing the full $P^t$ matrix. To the best of our knowledge, the size of the required $d$ to ensure a faithful distance preservation doesn't depend on how fast the eigenvalues decay to zero.
>
> ---
>
> **Convergence of $\widehat{\mathcal{H}}$ to $\mathcal{H}$.** We thank the reviewer for this comment, we added additional statistical properties of $\hat{\mathcal{H}}$ in the revised manuscript, (in particular, it's unbiasedness). Indeed we have a bound on the error that is in $O(N/\sqrt{n})$ with high probability, i.e. For any $\delta > 0$, with probability at least $1-\delta$:
> $$
>   |\hat{\mathcal{H}}(t) - \mathcal{H}(t)| \leq \frac{N}{\sqrt{n}} \frac{C(\pi_{(\nu)})}{2} \sqrt{\frac{N-n}{(N-1)\delta}}.
> $$
> where $C(\pi_{(\nu)})$ is the constant defined in Property 3.5, which might be defined component-wise for a multi-component digraph.
>
> ---
>
> **Minor comments.** We thank the reviewer for noticing these issues in the text, which will be fixed in the revised version.

---

> > ### Author Rebuttal · Reviewer_zxFJ · 2026-04-01
> >
> > Thank you for explaining the reasoning behind why ParPIC is preferable to existing approaches in certain (directed sparse) settings. I would encourage the authors to include this motivation clearly in the paper, as it is important in establishing the contribution and novelty.
> >
> > Regarding the diffusion time selection, naturally the aim of elbow heuristics is to avoid times which are "too large" or "too small"; my question is on why such a heuristic gives good results. Is there any theory to back up the claim that the elbow corresponds to a "good trade-off between noise and signal"? Or is this purely an empirical heuristic?
> >
> > Thank you for pointing me to the Johnson-Lindenstrauss lemma; this is indeed a useful way to think about the method. Do you have any explanation for why $d = \Theta(\log n)$ is too small in practice? Is this just a case of large hidden constants?
> >
> > The bound on the error of $\hat{\mathcal{H}}$ looks correct to me, and helps to provide a more quantitative analysis.
> >
> > I am adjusting my score to 5: Accept.

---

> > > ### Author Response · Authors · 2026-04-01
> > >
> > > We thank the reviewer once again for their comments and prompt responses. We are glad to see that the reviewer's concerns have been addressed and that they have updated their score to "accept".
> > >
> > > The motivation for using ParPIC instead of existing approaches will be included more clearly in the revised manuscript.
> > >
> > > **On the elbow heuristic.** The elbow heuristic is indeed an empirical method. To the best of our knowledge, there is no theoretical guarantee that the elbow corresponds to a "good trade-off between noise and signal". However, its long standing use in ML for detecting such trade-offs in various contexts (e.g. selecting the number of clusters in clustering, or selecting the number of dimensions in dimensionality reduction) suggests that it can be a useful tool for model (order/complexity) selection. The experimental results we report are aligned with this common ground in the field.
> > >
> > > **On Johnson-Lindenstrauss lemma.** Our understanding is that, indeed this is related to hidden constants, the lemma provides a bound of the form $d = O(\log(N)/\epsilon^2)$, where $\epsilon$ is the distortion level. As such, since we are dealing with clustering problems, distances need to be well-preserved to avoid misgrouping, leading to a requirement of a larger projection dimension, especially in the presence of clusters that are not well-separated. Using $\sqrt{N}$ is a "rule of thumb" that we found to work well in practice across all datasets, this allowed us to bypass the selection of $\epsilon$. We will add the $\log{N}$ case in Fig. 10 of the revised manuscript, and include a discussion on the theoretical guarantees provided by this lemma.

---

### Official Review · Reviewer_UM44 · 2026-03-01

**Soundness:** 3
**Presentation:** 3
**Significance:** 3
**Originality:** 3
**Overall Recommendation:** 5
**Confidence:** 3

**Summary:**

The work addresses the problem of clustering of directed graphs. The proposed method, ParPIC, builds on the Power-Iteration Clustering (PIC) paradigm and extends it to digraphs by introducing a parametrized reversible random walk (P-RW) operaton, depending on a vertex measure. It is designed as a convex combination of in- and out-degrees. The method is efficient since it avoids eigen-decomposition. An entropy-based criterion is used for the diffusion time selection. Experiments on synthetic and real-world digraphs show that ParPIC outperforms competing methods when directional structure is pronounced, while remaining competitive in near-undirected settings.

**Compliance With Llm Reviewing Policy:**

Affirmed.

**Final Justification:**

I thank the authors for the explanation. All my questions have been solved. I am adjusting my score to 5: Accept.

**Key Questions For Authors:**

1.	The paper jumps directly into the technical framework without first introducing or discussing what clustering means in the context of directed graphs. In undirected graphs, community structure is defined through edge density, but in digraphs the notion of a cluster is less obvious.
2.	Proposition 3.2 requires the undirected graph to be aperiodic for P(ν) to be aperiodic. Can the authors discuss how common this condition is in practice?
3.	The results are averaged over 100 independent runs, but it is not clear what varies across runs. Can the authors please explain this?
4.	I would ask the authors to improve Section 2. In Eq. (1), the stationary distribution π is used without being defined. Could the authors add a definition? Similarly, could they provide more intuition for the scale parameter t and explicitly introduce the eigenvalues λ and eigenvectors ϕ in Eq. (2) before using them? The paper states that "The terminology 'P is irreducible' or 'P is reversible' is used as shorthand for saying that the Markov chain has these properties", but never explicitly defines what these properties are.
5.	If the proposed P-RW operator and associated diffusion geometry are applied to an undirected graph, do they recover the standard definitions introduced in Section 2.2? Could the authors show explicitly under which choice of vertex measure ν the proposed framework reduces to classical diffusion geometry, as a sanity check for the generalization?
6.	AMI is a general clustering metric that measures agreement between a predicted partition and a ground truth, without accounting for the directed structure of the graph. Are there evaluation metrics more tailored to directed graph clustering? Could the authors discuss why AMI is the appropriate choice here, or complement it with a metric that better captures the quality of clustering with respect to edge directionality?
7.	Figure 10 reports the embedding dimension sensitivity for only 4 out of the 15 datasets used in the experiments. Could the authors provide results across all datasets, or justify the selection of these four.
8.	I would ask the authors to reference to the published version of a work instead of at its arXiv:

Kuchroo, Manik, et al. "Multimodal data visualization and denoising with integrated diffusion." IEEE International Workshop on Machine Learning for Signal Processing:[proceedings]. IEEE International Workshop on Machine Learning for Signal Processing. Vol. 2021. 2021.

Steinar Laenen, He Sun, Higher-Order Spectral Clustering of Directed Graphs, Advances in Neural Information Processing Systems 33, 2020.

B. Mohar, A new kind of Hermitian matrices for digraphs, Linear Algebra and its Applications, vol. 584, pp. 343–352, 2020, doi:10.1016/j.laa.2019.09.024.

Shan, Shan, and Ingrid Daubechies. "Diffusion maps: using the semigroup property for parameter tuning." Theoretical Physics, Wavelets, Analysis, Genomics: An Indisciplinary Tribute to Alex Grossmann. Cham: Springer International Publishing, 2022. 409-424.

**Limitations:**

yes

**Strengths And Weaknesses:**

Soundness.
The core ideas are sound. Proposition 3.2 establishes that the P-RW operator is reversible, ergodic, and irreducible under mild conditions, and the proofs in the appendix seem correct. The entropy-based time selection criterion is well-motivated. The authors also perform different ablation studies.
The aperiodicity condition in Proposition 3.2 requires the underlying undirected graph to be aperiodic, which may not hold in practice and is not discussed.

Presentation.
The paper pipeline is clearly illustrated in Figure 1.
The authors well explain the limitations of diffusion geometry applied to directed graphs.
Section 2 on diffusion geometry for undirected graphs is not easy to follow. Specifically, the stationary distribution π in Eq. (1) is never defined before its use, the scale parameter t lacks intuitive explanation, and the eigenvalues λ and eigenvectors ϕ in Eq. (2) are introduced without explicit definition.
Finally, what varies across runs is not clearly stated in the main text.

Significance.
Clustering directed graphs is a practically relevant and technically challenging problem, and the paper makes a meaningful contribution by extending the PIC framework to weakly connected digraphs in a principled way. The performance gains on intrinsically directed graphs with pronounced asymmetries are convincing and well-supported. The computational advantages over spectral methods are demonstrated empirically in Figure 3.

Originality.
The combination of a parametrized reversible random walk operator with the PIC framework for weakly connected digraphs is a novel and well-motivated contribution. The P-RW operator builds on recent works. The entropy-based, eigen-free time selection criterion is a practical and original contribution , extending prior eigen-based entropy measures to the eigen-free setting.  The low-dimensional random projection approximation is less novel.
Overall, the main contribution lies in the combination and integration of these components into a coherent framework for weakly connected digraphs.

---

> ### Author Rebuttal · Authors · 2026-03-30
>
> We would like to thank the reviewer for their positive and constructive feedback. We address the main comments below, and will make sure to include the suggested improvements in the revised version of the paper.
>
> **Digraph clustering.** To increase clarity, we have added a paragraph to the introduction to better motivate what clustering means in the digraph setting. In undirected graphs, clustered vertices are more densely connected internally, and random walks starting from those would exhibit similar behaviors. Defining clusters in digraphs is more complex due to directionality. Some methods use flow-based definitions (Cucuringu et al. 2020) or symmetrization-based (Satuluri & Parthasarathy, 2011), but these may lose important directional information. Our approach defines clusters as sets of vertices with similar $t$-step transition distributions $P_{(\nu)}^t(i, \cdot)$, capturing both metastability and stochastic equivalence (as in DiSBM, where vertices in the same block have identical transition distributions in expectation).
>
> ---
>
> **On the aperiodicity hypothesis.** In practice this condition is not very restrictive as 'most' real-world graphs are aperiodic; when the graph is periodic, one can always add a small amount of self-loops to break this periodicity without significantly altering the graph structure nor the sparsity of the matrix.
>
> ---
>
> **On the averaging in experiments.** In the experiments, what varies is the random initialization of the k-means algorithm, as well as the random projection used for the low-dimensional embedding. We will clarify this in the revised version.
>
> ---
>
> **On Sec. 2.** We thank the reviewer for these remarks, which allowed us to improve the presentation of the paper. We have added definitions for the stationary distribution, the scale parameter, and irreducibility, aperiodicity and reversibility. Intuitively, the scale parameter $t$ is the diffusion time: $\mathbf{P}^t(i,j)$ gives the probability of reaching $j$ from $i$ in $t$ steps. For large $t$, $\mathbf{P}^t$ converges to a rank-one matrix determined by the stationary distribution $\pi$, when it exists.
>
> ---
>
> **Using the P-RW operator on undirected graphs.** Indeed, the proposed P-RW operator and the associated diffusion geometry do recover the standard definitions when applied to an undirected graph. This can be seen using $\nu = \pi$, in this case $\xi = \pi$ and, due to the detailed balance equation, $D_\pi P = P^\top D_\pi$. This will be added to Prop. 3.2.
>
> ---
>
> **On AMI.**  AMI is a standard metric for clustering evaluation, and is widely used in the literature. The fundamental assumption of graph-based clustering is that there is a correlation between true vertex labels and the way vertices are connected together within the graph. Thus, examining the graph structure (regardless if it is undirected or directed) can lead to a label-wise meaningful graph partitions. In that sense, AMI is a relevant metric to evaluate the quality of vertex clustering. We are not aware of any supervised metric that would be tailored to vertex clustering in digraphs.
>
> ---
>
> **Fig. 10 and embedding dim sensitivity.** These four datasets were selected as they are representative of the different types of datasets used in the experiments: Iris and WDBC are representative of the $K$-NN constructed graphs while Email Eu and PolBlogs are representative for the network-based datasets. Generally, the behavior of sensitivity with respect to the embedding dimension is similar across all dataset, with $\sqrt{N}$ being sufficient to achieve good performance. We will include results on additional datasets in the revised manuscript as to support this claim. Additional informations on the embedding dimension justifications can be seen in answers to Rev. 8str.
>
> ---
>
> **Typos/Published works.** The citations have been fixed for the revised version.

---

> > ### Author Rebuttal · Reviewer_UM44 · 2026-04-01
> >
> > I thank the authors for the explanation. All my questions have been solved.
> > I am adjusting my score to 5: Accept.

---

> > > ### Author Response · Authors · 2026-04-03
> > >
> > > We thank the reviewer once again for their comments and prompt responses. We are pleased that our revisions have addressed their concerns and appreciate their decision to update the score to “accept.”

---

### Official Review · Reviewer_8str · 2026-03-09

**Soundness:** 3
**Presentation:** 3
**Significance:** 3
**Originality:** 3
**Overall Recommendation:** 4
**Confidence:** 4

**Summary:**

This paper proposes a method for clustering directed graphs. Given a random walk operator P associated to a digraph with N vertices, it provides an associated reversible transition matrix P_\nu, that is parametrized by a vector of node weights \nu=(\nu_i). It then proposes to cluster vertices i based on their embeddings ((P_\nu)^t(i,j), j\in N]), where t is a suitable time horizon. The paper further proposes an entropy criterion for determining a suitable time horizon t. It also proposes to use a lower-dimensional embedding in dimension d<N, based on the rows of (P_\nu)^t Z, where Z is a random N times d matrix. To further reduce the computational load, the paper proposes a sampling-based approximation of the entropy measure used to determine the time horizon t.

The paper then provides experimental comparison of the proposed method with prior approaches. The proposed method compares favorably with competitors, especially on datasets which display strong directionality or heterogeneity in degrees.

**Compliance With Llm Reviewing Policy:**

Affirmed.

**Key Questions For Authors:**

Has the entropic criterion been proposed in earlier litterature? If so, a citation is needed; if not, it could be more strongly advertised as a contribution.

It is stated at several places that the constructed matrix P_\nu preserves directionality, but it is not explained in detail in what sense this holds. Some comments on this would be helpful.

There appears to be a problem in Proposition 3.5, monotonicity of H(t): your proof assumes that the transpose of P_\nu is stochastic, which is not necessarily the case. Also, Figure 8 shows that entropy may not increase monotonically. Instead, you could rely on monotonicity of Kullback-Leibler divergence between distribution at t and stationary distribution, a classical fact established in Cover and Thomas, Elements of Information Theory.

Could you provide some justification for the low-dimensional embedding used, for instance exploiting the results of Tropp and Weber, Randomized algorithms for low-rank matrix approximation: Design, analysis, and applications?

What version of k-means is used for producing the final clusters?

Typos: p2, lines 99 and 100: extra ().
line 118: non-unicity --> non-uniqueness.

**Limitations:**

yes

**Strengths And Weaknesses:**

The paper's proposal seems well-motivated, and contains a number of interesting ideas. The experimental validations appear thorough. The presentation is satisfactory, the paper reading quite well. The task of clustering graph data is important, so that the work is well-motivated. The contribution's significance might be strengthened by providing an in-depth discussion of the scalability of the approach for larger datasets. The entropic criterion for choosing the seems interesting. All in all, the paper provides a nice combination of ideas, improving on the method in [Sevi et al] by making it eigenanalysis-free, and incorporating a number of relevant ingredients such as dimensionality reduction via random embedding, and entropy criterion for selection of time horizon. It would have been stronger if the various ingredients used (entropic criterion, random matrix for reducing the dimensionality of matrix rows to be embedded) had been given some theoretical justification.

---

> ### Author Rebuttal · Authors · 2026-03-30
>
> We would like to thank the reviewers for their positive and constructive feedback. We appreciate the recognition of the usefulness of the proposed entropic criterion. We address the main comments below, and will make sure to include all the suggested improvements in the revised version of the paper.
>
> **On the entropic criterion.** We thank the reviewer for this suggestion. The entropic criterion is indeed a novel contribution of our work, and we agree that it could be better highlighted as such. While it is inspired by recent diffusion-based methods ([Debaussart-Joniec & Kalogeratos, 2026](https://arxiv.org/pdf/2512.01484); [Kuchroo et al., 2021](https://icml-compbio.github.io/2021/papers/WCBICML2021_paper_35.pdf)), those approaches rely on the SVD or eigen-decomposition of the random-walk operator. In contrast, our entropic criterion is fully eigen-free and leverages the iterated matrix directly. This brings both conceptual and computational advantages, notably through the use of the approach to approximate its value using random samples, which might be easily parallelized. We will make sure to emphasize this point more clearly in the revised manuscript.
>
> ---
>
> **Directionality preservation.** The P-RW operator $P_{(\nu)}$ preserves directionality by using the vertex measure $\nu$ to weight forward and backward transitions, ensuring that directed information is retained. Its stationary distribution $\pi_{(\nu)} \propto \nu + \xi$ (with $\xi = \mathbf{P}^\top \nu$) highlights important vertices according to both $\nu$ and it's pushed version $\xi$, which depends on the directionality. Unlike naive symmetrization, which averages the adjacency matrix and loses directionality, or teleportation-based methods, which destroy sparsity, the P-RW operator maintains both directionality and sparsity. This enables it to distinguish core-perhiphery edges from core-core or periphery-periphery edges, something symmetrization cannot achieve.
>
> ---
>
> **Prop 3.5 and Fig.8.** We thank the reviewer for spotting this issue with Fig.8. The observed non-monotonicity was because we had mistakenly inserted a wrong plots, affecting only this figure. We apologize for this oversight; plots will be corrected in the revised version and are accessible on anonymous.4open.science [here](https://anonymous.4open.science/r/parpic_images-06FA/entropy_figures/updated_fig8.png). With this correction, we confirm that the entropy is non-increasing when computed with the P-RW operator, even for graphs with multiple connected components. This clarification makes it unnecessary to invoke the Kullback-Leibler divergence. Finally, Proposition 3.5 does not require $P_{(\nu)}^\top$ to be stochastic; it is enough that $P_{(\nu)}^t$ is stochastic for all $t \ge 1$, which follows from $P_{(\nu)}$ being stochastic. Indeed, the proof relies on the fact that $P^{t+1}(i,j) = \sum_k P^{t}(i,k) P(k,j)$, and the fact that the entropy can be written as $H_i (t) = \sum_j f(P^t(i,j))$, where $f:x \mapsto - x \log{x}$ is concave. Using this we can apply Jensen's inequality to have:
> $$
> H_i (t+1) = \sum_j f \left(\sum_k P^t(i,k) P(k,j)\right) \ge \sum_{k} f(P^t(i,k)) \sum_j P(k,j) = H_i (t).
> $$
> We will also clarify in the revised manuscript that $P^t$ denotes the $t$-th power, not the transpose.
>
> ---
>
> **On the low-dim embedding.** We thank the reviewer for this suggestion and useful reference. Our use of random projection is motivated by the Johnson–Lindenstrauss lemma. Since the objective is to cluster the rows of $P_{(\nu)}$, the lemma guarantees that pairwise distances between these rows are approximately preserved under a suitable random projection. In particular, for the projected matrix $Z_t$ we have:
> $$
>  || Z_t(i, \cdot) - Z_t(j, \cdot) || \approx || P^t(i, \cdot) - P^t(j, \cdot) ||
> $$
> with high probability, provided the projection dimension is sufficiently large. In practice, we found that a projection dimension on the order of $\sqrt{N}$ works well, likely due to the relatively small size of the datasets considered. From a theoretical standpoint, however, a dimension on the order of $O(\log{N})$ (up to a multiplicative constant depending on the desired accuracy) is sufficient to ensure the JL guarantee. This allows us to efficiently embed the data in a lower-dimensional space without explicitly computing the full $P^t$ matrix, by repetitive actions of $P$ on $Z$. The proposed approach differs from the methods studied in Tropp and Webber (2023) as those methods focus on approximating the leading eigen/singular-vectors of a matrix, while our approach directly approximates distances of the rows of $P^t$ without relying on spectral properties. We will clarify this point in the revised manuscript and include references to the relevant literature.
>
> ---
>
> **On K-means.** We use the standard Lloyd's algorithm for k-means++, as included in `sklearn.cluster`.
>
> ---
>
> **Typos.** We thank the reviewer for noticing these typos, which have been fixed in the revised version.

---

> > ### Author Rebuttal · Reviewer_8str · 2026-04-03
> >
> > The authors' answers are satisfactory, and there is no need for further discussion to address my questions / concerns.

---

> > > ### Author Response · Authors · 2026-04-03
> > >
> > > We thank the reviewer once again for their constructive comments. We are glad that the concerns have been satisfactorily addressed.

---

### Official Review · Reviewer_EaAX · 2026-03-12

**Soundness:** 3
**Presentation:** 4
**Significance:** 3
**Originality:** 3
**Overall Recommendation:** 5
**Confidence:** 3

**Summary:**

This paper studies the problem of vertex clustering in directed graphs, where classical diffusion geometry and spectral methods are mainly designed for undirected graphs and are not directly applicable. The authors construct a parametrized reversible random walk operator by designing vertex measures to enable diffusion-based analysis on digraphs. An entropy-based criterion is further introduced to select an appropriate diffusion time. To improve scalability, the method adopts a PIC-style algorithm that avoids explicit eigen-decomposition, thereby reducing memory requirements and computational cost. The resulting low-dimensional representations are then used for clustering to achieve vertex-level clustering on directed graphs.

**Compliance With Llm Reviewing Policy:**

Affirmed.

**Final Justification:**

This paper presents a flexible parametrized formulation that enables the diffusion process to adapt to different directed graph structures. The theoretical analysis, particularly the continuity of the P-RW operator with respect to the vertex measure, provides important guarantees on the stability of the method.

I appreciate the authors’ thorough rebuttal, which addresses my concerns and further clarifies the theoretical aspects of the work. The responses reinforce my positive assessment of the paper.

**Key Questions For Authors:**

1、The paper lacks evidence of scalability to large graphs. Could the authors clarify how the proposed ParPIC method is expected to perform on larger graphs?

2、The authors design a vertex measure as a convex combination of in-degree and out-degree, controlled by a hyperparameter γ. Could the authors clarify how γ should be chosen in different real-world scenarios? Is there a principled strategy for its selection, and can Figure 6 provide further insights into the graph structure that explains why the model’s performance may favor in-degree or out-degree in certain datasets?

3、As shown in Figure 7 (e, g, h), the model’s performance exhibits significant fluctuations with respect to the diffusion time, indicating sensitivity. Could the authors clarify whether this instability is related to the design of the vertex measure, the intrinsic properties of the datasets, or other factors? Would these fluctuations affect the model’s ability to consistently achieve optimal performance?

**Limitations:**

Yes

**Strengths And Weaknesses:**

Strength:

1、The proposed parametrized formulation provides additional flexibility, allowing the diffusion process to adapt to different directed graph structures through the choice of the vertex measure.

2、The paper provides theoretical analysis showing that the proposed P-RW operator is continuous with respect to the vertex measure, ensuring stability of the operator under small perturbations of the measure.

Weaknesses:

1、The model’s performance depends on the choice of the vertex measure parameter (γ), and there is no adaptive strategy to select an optimal value in new or unseen graph scenarios.

2、The datasets are all small to moderate in size (max N=4000), which limits the evidence for the scalability of the proposed method.

---

> ### Author Rebuttal · Authors · 2026-03-30
>
> We would like to thank the reviewers for their positive and constructive feedback. We address the main comments below, and will make sure to include the suggested improvements in the revised version of the paper.
>
> **Scalability.** ParPIC is designed to scale efficiently to large graphs, as shown in Fig. 3, which demonstrates low computational complexity compared to standard spectral methods. This scalability is supported by the method being fully eigen-free and relying on matrix-vector products, which are efficient for sparse matrices and can be parallelized. Empirically, the chosen diffusion times yield strong performance across all datasets while being small enough to be computationally advantageous (see Fig. 7).
>
> We will include a more detailed complexity analysis in the revised paper.
> Moreover, we will extend Fig. 3 to demonstrate scalability in larger graphs. Results can be seen on anonymous.4open.science  [here](https://anonymous.4open.science/r/parpic_images-06FA/scaling_runtime.pdf): for a graph of 15k vertices, Spectral Clustering (SC) cannot be computed in under 2 minutes, ParPIC (full) needs ~66 secs, and ParPIC (low-dim) needs ~5 secs, while on 30k vertices it needs ~18 secs. Since most existing methods for digraph clustering are spectral-based and do not scale well, the available benchmark datasets are relatively small.
>
> ---
>
> **Choice for gamma.** The parameter $\gamma$ controls the relative weight of in- and out-degree in the vertex measure, which in turn shapes the stationary distribution $\pi_{(\nu)} \propto \nu + \xi$ and hence the diffusion geometry. The patterns in Figure 6 have concrete structural explanations that allow us to provide the baseline $\gamma = 0.5$. For $K$-NN graphs, performance is generally flat, with a decay when $\gamma$ approaches $0$ or $1$, this is explained by the uniform out-degree of $K$-NN graphs, implying that in this case the vertex-measure $\nu = \gamma d_\textnormal{in} + (1-\gamma) \mathbf{1}/N$. On network datasets, a similar behavior is observed, where extremal values of $\gamma$ can lead to worse performance. This implies that setting $\gamma = 0.5$ is a good default choice, as it allows to balance the contribution of in-degree and out-degree, and thus to capture both sources and sinks in the graph.
>
> ---
>
> **Diffusion time sensitivity.** The time $t$ is indeed a key parameter of the method.
> Its influence can be understood via the eigen-decomposition of the transition matrix $\mathbf{P}$: $\mathbf{P}^t = \sum_{k=1}^N \lambda_k^t \phi_k \psi_k^\top$. As $t$ increases, contributions from smaller eigenvalues decay, and the dynamics get dominated by the leading eigenvectors, which may not always capture sufficiently the graph structure. If $t$ is too small, noise dominates. Thus, careful selection of $t$ is important to balance signal and noise.
>
> While the vertex measure indeed impacts the properties of the P-RW operator, the sensitivity to time is inherent to all diffusion-based methods, and its selection is a common problem (see, e.g. [Shan & Daubechies, 2022](https://arxiv.org/abs/2203.02867)) that is not specific to using a parametrized vertex measure, using such a measure does not incur additional sensitivity to the diffusion time.

---

> > ### Author Rebuttal · Reviewer_EaAX · 2026-04-03
> >
> > Thanks for your detailed response. My questions are well addressed.

---

> > > ### Author Response · Authors · 2026-04-03
> > >
> > > We thank the reviewer for their encouraging feedback and for acknowledging our clarifications. We are glad that our revisions addressed the concerns raised.

---

### Decision · Program_Chairs · 2026-04-30

**Decision:**

Accept (regular)

**Comment:**

This paper studies the problem of vertex clustering in directed graphs, where typical spectral methods are mainly designed for undirected graphs and are not directly applicable. All the reviewers gave positive evaluation on the paper, and I also believe that the paper will make a valuable contribution to the program of ICML 2026.